# Understanding Mode Connectivity via Parameter Space Symmetry

Bo Zhao [1]   Nima Dehmamy [2]   Robin Walters [3]   Rose Yu [1]

## Abstract

Neural network minima are often connected by curves along which train and test loss remain nearly constant, a phenomenon known as mode connectivity. While this property has enabled applications such as model merging and fine-tuning, its theoretical explanation remains unclear. We propose a new approach to exploring the connectedness of minima using parameter space symmetry. By linking the topology of symmetry groups to that of the minima, we derive the number of connected components of the minima of linear networks and show that skip connections reduce this number. We then examine when mode connectivity and linear mode connectivity hold or fail, using parameter symmetries which account for a significant part of the minimum. Finally, we provide explicit expressions for connecting curves in the minima induced by symmetry. Using the curvature of these curves, we derive conditions under which linear mode connectivity approximately holds. Our findings highlight the role of continuous symmetries in understanding the neural network loss landscape.

## 1. Introduction

Among recent studies on the loss landscape, a particularly interesting finding is mode connectivity (Draxler et al., 2018; Garipov et al., 2018)—the observation that distinct minima found by stochastic gradient descent (SGD) can be connected by continuous, low-loss paths through the high-dimensional parameter space. Mode connectivity has important implications for other aspects of deep learning theory, including the lottery ticket hypothesis (Frankle et al., 2020) and the analysis of loss landscapes and training trajectories

[1]University of California, San Diego  [2]IBM Research  [3]Northeastern University. Correspondence to: Bo Zhao <bozhao@ucsd.edu>, Nima Dehmamy <nima.dehmamy@ibm.com>, Robin Walters <r.walters@northeastern.edu>, Rose Yu <roseyu@ucsd.edu>.

*Proceedings of the 42nd International Conference on Machine Learning*, Vancouver, Canada. PMLR 267, 2025. Copyright 2025 by the author(s).

(Gotmare et al., 2018). Mode connectivity has also inspired applications in diverse fields, including model ensembling (Garipov et al., 2018; Benton et al., 2021; Benzing et al., 2022), model averaging (Izmailov et al., 2018; Wortsman et al., 2022), pruning (Frankle et al., 2020), improving adversarial robustness (Zhao et al., 2020), and fine-tuning for altering prediction mechanism (Lubana et al., 2023).

Despite extensive empirical validation, mode connectivity, especially linear mode connectivity, remains largely a theoretical conjecture (Altintas et al., 2023). The limited theoretical explanation suggests a need for new proof techniques. In this paper, we focus on parameter symmetries, which encode information about the structure of the parameter space and the minimum. Our work introduces a new approach towards understanding the topology of the minimum and complements existing theories on mode connectivity (Yunis et al., 2022; Freeman & Bruna, 2017; Nguyen, 2019; 2021; Kuditipudi et al., 2019; Shevchenko & Mondelli, 2020; Nguyen et al., 2021).

Discrete symmetry is known to be related to mode connectivity. In particular, the neural network output, and hence the minimum, is invariant under neuron permutations (Hecht-Nielsen, 1990). Several algorithms have been developed to find optimal permutations for linear connectivity (Singh & Jaggi, 2020; Ainsworth et al., 2023), and Entezari et al. (2022) conjecture that all minima found by SGD are linearly connected up to permutation. Compared to discrete symmetry, the role of continuous symmetry, such as positive rescaling in ReLU, in shaping loss landscape remains less well studied.

We explore the connectedness of minimum through continuous symmetries in the parameter space. Continuous symmetry groups with continuous actions define positive dimensional connected spaces in the minimum (Zhao et al., 2023). By relating topological properties of symmetry groups to their orbits and the minimum, we show that both continuous and discrete symmetry are useful in understanding the origin and failure cases of mode connectivity. Additionally, continuous symmetry defines curves on the minimum (Zhao et al., 2024). This enables a principled method for deriving explicit expressions for paths connecting two minima, a task that previously relied on empirical approaches.

Our main contributions are:

- Providing the number of connected components of full-rank linear regression with and without skip connections, by relating topological properties of symmetry groups to those of minima.

- Proving mode connectivity up to permutation for linear networks with invertible weights.

- Deriving examples where the error barrier on linear interpolation of minima is unbounded.

- Deriving explicit low-loss curves that connect minima related by symmetry, and bounding the loss barrier on linear interpolations between minima using the curvature of these curves.

## 2. Related Work

**Mode connectivity.** Garipov et al. (2018) and Draxler et al. (2018) discover empirically that the minima of neural networks are connected by curves on which train and test loss are almost constant. It is then observed that SGD solutions are linearly connected if they are trained from pre-trained weights (Neyshabur et al., 2020) or share a short period of training at the beginning (Frankle et al., 2020). Additionally, neuron alignment by permutation improves mode connectivity (Singh & Jaggi, 2020; Tatro et al., 2020). Subsequently, Entezari et al. (2022) conjecture that all minima found by SGD are linearly connected up to permutation. Following the conjecture, Ainsworth et al. (2023) develop algorithms that find the optimal alignment for linear mode connectivity, and Jordan et al. (2023) further reduce the barrier by rescaling the preactivations of interpolated networks.

It is worth noting that linear mode connectivity does not always hold outside of computer vision. Language models that are not linearly connected have different generalization strategies (Juneja et al., 2023). Lubana et al. (2023) further show that the lack of linear connectivity indicates that the two models rely on different attributes to make predictions. We derive new theoretical examples of failure cases of linear mode connectivity (Section 5.2).

**Theory on connectedness of minimum.** Several work explores the theoretical explanation of mode connectivity by studying the connectedness of sub-level sets. Freeman & Bruna (2017) show that the minimum is connected for 2-layer linear network without regularization, and for deeper linear networks with $L2$ regularization. Futhermore, they show that the minimum of a two-layer ReLU network is asymptotically connected, that is, there exists a path connecting any two solutions with bounded error. Nguyen (2019) proves that the sublevel sets are connected in pyramidal networks with piecewise linear activation functions and first hidden layer wider than $2N$, where $N$ is the number of

training data). The width requirement is later improved to $N + 1$ (Nguyen, 2021).

Others prove connectivity under dropout stability. Kuditipudi et al. (2019) show that a piece-wise linear path exists between two solutions of ReLU networks, if they are both dropout stable, or both noise stable and sufficiently over-parametrized. Shevchenko & Mondelli (2020) generalize this proof to show that wider neural networks are more connected, following the observation that SGD solutions for wider network are more dropout stable. Nguyen et al. (2021) give a new upper bound of the loss barrier between solutions using the loss of sparse subnetworks that are optimized, which is a milder condition than dropout stability.

A few papers provide theoretical insights into linear mode connectivity using different approaches. Yunis et al. (2022) explain linear mode connectivity by finding a convex hull defined by SGD trajectory endpoints. Ferbach et al. (2023) use optimal transport theory to prove that wide two-layer neural networks trained with SGD are linearly connected with high probability. (Singh et al., 2024) explain the topography of the loss landscape that enables or obstructs linear mode connectivity. Zhou et al. (2023) show that the feature maps of each layer are also linearly connected and identify conditions that guarantee linear connectivity. Altintas et al. (2023) analyze effects of architecture, optimization algorithm, and dataset on linear mode connectivity empirically.

We approach the theoretical origin of mode connectivity via continuous symmetries in the parameter space, a connection that has not been previously established. This connection leads to new topological results and explicit expressions of low loss curves. Using these results, we also contribute to the understanding for linear mode connectivity by providing conditions under which it approximately holds.

**Symmetry in the loss landscape.** Discrete symmetries have inspired a line of work on loss landscapes. Brea et al. (2019) show that permutations of a layer are connected within a loss level set. By analyzing permutation symmetries, Şimşek et al. (2021) characterize the geometry of the global minima manifold for networks and show that adding one neuron to each layer in a minimal network connects the permutation equivalent global minima. Continuous symmetries have also gained attention in optimization (Badrinarayanan et al., 2015; Petzka et al., 2020; Kunin et al., 2021; Zhao et al., 2022). By removing permutation and rescaling symmetries, Pittorino et al. (2022) study the geometry of minima in the functional space. Zhao et al. (2023) find a set of nonlinear continuous symmetries that partially parametrizes the minimum. Zhao et al. (2024) use symmetry induced curves to approximate the curvature of the minimum. Our paper explores a new application of parameter symmetry—explaining the connectedness of the minimum.

# 3. Preliminaries

In this section, we review mathematical concepts used in the paper and list some useful results on the number of connected components of topological spaces. A more detailed version with proofs can be found in Appendix A.

## 3.1. Connected components

Consider two topological spaces $X$ and $Y$. A map $f : X \to Y$ is *continuous* if for every open subset $U \subseteq Y$, its preimage $f^{-1}(U)$ is open in $X$. If $X$ and $Y$ are metric spaces with metrics $d_X$ and $d_Y$ respectively, this is equivalent to the delta-epsilon definition. That is, $f$ is continuous if at every $x \in X$, for any $\epsilon > 0$ there exists $\delta > 0$ such that $d_X(x, y) < \delta$ implies $d_Y(f(x), f(y)) < \epsilon$ for all $y \in X$.

A topological space is *connected* if it cannot be expressed as the union of two disjoint, nonempty, open subsets. A topological space $X$ is *path connected* if for every $p, q \in X$, there is a continuous map $f : [0, 1] \to X$ such that $f(0) = p$ and $f(1) = q$. Path connectedness implies connectedness. The converse is not always true (Lee, 2010), but counterexamples are often specifically constructed and unlikely to be encountered in the context of deep learning. Path connectedness can therefore help develop intuition for connectedness, for practical purposes.

The following theorem is the main intuition of this paper and will appear frequently in proofs.

**Theorem 3.1** (Theorem 4.7 in (Lee, 2010)). *Let $X, Y$ be topological spaces and let $f : X \to Y$ be a continuous map. If $X$ is connected, then $f(X)$ is connected.*

A map $f$ is a *homeomorphism* from $X$ to $Y$ if $f$ is bijective and both $f$ and $f^{-1}$ are continuous. $X$ and $Y$ are *homeomorphic* if such a map exists. A *(connected) component* of a topological space $X$ is a maximal nonempty connected subset of $X$. The components of $X$ form a partition of $X$. The next two corollaries of Theorem 3.1 show that connectedness and the number of connected components are topological properties. That is, they are preserved under homeomorphisms.

**Corollary 3.2.** *Let $f : X \to Y$ be a homeomorphism from $X$ to $Y$, and let $U \subseteq X$ be a subset of $X$ with the subspace topology. Then $U$ is connected if and only if $f(U) \subseteq Y$ is connected.*

**Corollary 3.3.** *Let $X$ be a topological space that has $N$ components. Let $Y$ be a topological space homeomorphic to $X$. Then $Y$ has $N$ components.*

Another consequence of Theorem 3.1 is the following upper bound on the number of components of the image of a continuous map.

**Proposition 3.4.** *Let $f : X \to Y$ be a continuous map. The number of components of the image $f(X) \subseteq Y$ is at most the number of components of $X$.*

Let $X_1, ..., X_n$ be topological spaces. The *product space* is their Cartesian product $X_1 \times ... \times X_n$ endowed with the product topology. Denote $\pi_0(X)$ as the set of connected components of a space $X$. The following proposition provides a way to count the components of a product space.

**Proposition 3.5.** *Consider $n$ topological spaces $X_1, ..., X_n$. Then $|\pi_0(X_1 \times ... \times X_n)| = \prod_{i=0}^{n} |\pi_0(X_i)|$.*

## 3.2. Groups

A *group* is a set $G$ together with a composition law, written as juxtaposition, that satisfies associativity, $(ab)c = a(bc)$ $\forall\, a, b, c \in G$, has an identity $1$ such that $1a = a1 = a$ $\forall\, a \in G$, and for all $a \in G$, there exists an inverse $b$ such that $ab = ba = 1$. An *action* of a group $G$ on a set $S$ is a map $\cdot : G \times S \to S$ that satisfies $1 \cdot s = s$ for all $s \in S$ and $(gg') \cdot s = g \cdot (g' \cdot s)$ for all $g, g'$ in $G$ and all $s$ in $S$. The *orbit* of $s \in S$ is the set $O(s) = \{s' \in S \mid s' = gs \text{ for some } g \in G\}$.

A *topological group* is a group $G$ endowed with a topology such that multiplication and inverse are both continuous. A recurring example is the general linear group $GL_n(\mathbb{R})$, with the subspace topology obtained from $\mathbb{R}^{n^2}$. The group $GL_n(\mathbb{R})$ has two connected components, which correspond to matrices with positive and negative determinant.

The *product* of groups $G_1, ..., G_n$ is a group denoted by $G_1 \times ... \times G_n$. The set underlying $G_1 \times ... \times G_n$ is the Cartesian product of $G_1, ..., G_n$. The group structure is defined by identity $(1, ..., 1)$, inverse $(g_1, ..., g_n)^{-1} = (g_1^{-1}, ..., g_n^{-1})$, and multiplication rule $(g_1, ..., g_n)(g_1', ..., g_n') = (g_1 g_1', ..., g_n g_n')$.

## 3.3. Connectedness of groups, orbits, and level sets

From Theorem 3.1, continuous maps preserve connectedness. Through continuous actions, we study the connectedness of orbits and level sets by relating them to the connectedness of more familiar objects such as the general linear group. Establishing a homeomorphism from the group to the set of minima requires the symmetry group's action to be continuous, transitive, and free. Here we only assume the action to be continuous and try to bound the number of components of the orbits.

As an immediate consequence of Proposition 3.4, an orbit cannot have more components than the group.

**Corollary 3.6.** *Assume that the action of a group $G$ on $S$ is continuous. Then the number of connected components of orbit $O(s)$ is smaller than or equal to the number of connected components of $G$, for all $s$ in $S$.*

Let $X$ be a topological space and $L : X \to \mathbb{R}$ a continuous function on $X$. A topological group $G$ is said to be a *symmetry group* of $L$ if $L(g \cdot x) = L(x)$ for all $g \in G$ and $x \in X$. In this case, the action can be defined on a level set of $L$, $L^{-1}(c)$ with a $c \in \mathbb{R}$, as $G \times L^{-1}(c) \to L^{-1}(c)$. If the minimum of $L$ consists of a single orbit, Corollary 3.6 extends to the number of components of the minimum.

**Corollary 3.7.** *Let $L$ be a function with a symmetry group $G$. If the minimum of $L$ consists of a single $G$-orbit, then the number of connected components of the minimum is smaller or equal to the number of connected components of $G$.*

Generally, symmetry groups do not act transitively on a level set $L^{-1}(c) \in X$. In this case, the connectedness of the orbits does not directly inform the connectedness of the level set. Nevertheless, since the set of orbits partitions the space, we can use the following bound on the number of components of the space.

**Proposition 3.8.** *Let $X$ be a topological space and let $X = \coprod_i X_i$ be a partition of $X$ into disjoint subspaces. Then $|\pi_0(X)| \leq \sum_i |\pi_0(X_i)|$.*

Consider a topological space $X$ and a group $G$ that acts on $X$. Let $O = \{O_1, ..., O_n\}$ be the set of orbits. By Proposition 3.8, the number of components of the orbits give the following upper bound on the number of components of the space: $|\pi_0(X)| \leq \sum_{i=1}^n |\pi_0(O_i)|$.

## 4. Connected Components of the Minimum

In this section, we relate topological properties of symmetry groups to topological properties of the minimum. In particular, we provide the number of connected components of the minimum when all symmetries are known. Omitted proofs can be found in Appendix C.

### 4.1. Linear network with invertible weights

Let Param be the space of parameters. Consider the multilayer loss function $L :$ Param $\to \mathbb{R}$,

$$L : \text{Param} \to \mathbb{R}, \quad (W_1, ..., W_l) \mapsto ||Y - W_l...W_1X||_2^2. \quad (1)$$

where $X, Y \in \mathbb{R}^{h \times h}$ are the input and output of the network. In this subsection, we assume that both $X, Y$ have rank $h$, and Param $= (\mathbb{R}^{h \times h})^l$. Then $L$ is invariant to $GL_h(\mathbb{R})^{l-1}$, which acts on Param by $g \cdot (W_1, ..., W_l) = (g_1 W_1, g_2 W_2 g_1^{-1}, ..., g_{l-1} W_{l-1} g_{l-2}^{-1}, W_l g_{l-1}^{-1})$, for $(g_1, ..., g_{l-1}) \in GL_h(\mathbb{R})^{l-1}$.

Let $L^{-1}(c) = \{\theta \in \text{Param} : L(\theta) = c\}$ be a level set of $L$. Since $|| \cdot ||_2 \geq 0$ and $L^{-1}(0) \neq \emptyset$, the minimum value of $L$ is 0. By relating the topology of $GL_h(\mathbb{R})$ and $L^{-1}(0)$, we have the following observations on the structure of the minimum of $L$.

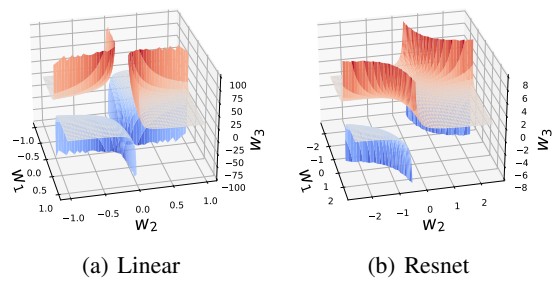

(a) Linear        (b) Resnet

Figure 1: Minimum of (a) 3-layer linear net $||Y - W_3 W_2 W_1 X||_2$ and (b) 3-layer linear net with a residual connection $||Y - W_3(W_2 W_1 X + X)||_2$, where $X = 1$, $Y = 1$, and $W_1, W_2, W_3 \in \mathbb{R}$.

**Proposition 4.1.** *There is a homeomorphism between $L^{-1}(0)$ and $(\text{GL}_h)^{l-1}$.*

Since $(\text{GL}_h)^{l-1}$ has $2^{l-1}$ connected components and homeomorphisms preserve topological properties, $L^{-1}(0)$ also has $2^{l-1}$ connected components. Note that this number is independent of the network width, due to the fact that $GL_n(\mathbb{R})$) has two connected components regardless of $n$.

**Corollary 4.2.** *The minimum of $L$ has $2^{l-1}$ connected components.*

### 4.2. ResNet with 1D weights

The topological properties of the minimum set depend on the architecture. As an example of this dependency, we show that adding a skip connection changes the number of connected components of the minimum.

Consider a residual network $W_3(W_2 W_1 X + \varepsilon X)$ and loss function

$$L(W_3, W_2, W_1) = ||Y - W_3(W_2 W_1 X + \varepsilon X)||_2, \quad (2)$$

where $(W_1, W_2, W_3) \in \text{Param} = \mathbb{R}^{n \times n} \times \mathbb{R}^{n \times n} \times \mathbb{R}^{n \times n}$, $\varepsilon \in \mathbb{R}$, and data $X \in \mathbb{R}^{n \times n}, Y \in R^{n \times n}$. The following proposition states that for a three-layer residual network with weight matrices of dimension $1 \times 1$, the number of components of the minimum is smaller than that of a linear network without the skip connection.

**Proposition 4.3.** *Let $n = 1$. Assume that $X, Y \neq 0$. When $\varepsilon = 0$, the minimum of $L$ has 4 connected components. When $\varepsilon \neq 0$, the minimum of $L$ has 3 connected components.*

The $\varepsilon = 0$ case follows from Corollary 4.2. For the $\varepsilon \neq 0$ case, the proof decomposes the minimum of $L$ into two sets $S_1$ and $S_0$, corresponding to the minima without the skip connection and an extra set of solutions because of the skip connection. $S_1$ is homeomorphic to $GL_1 \times GL_1$ and has

4 connected components. $S_0$ is a line and has 1 connected component. Two components of $S_1$ are connected to $S_0$, while the other two components of $S_1$ are not. Therefore, $S_0$ connects two components of $S_1$. As a result, the minimum of $L$ has 3 connected components.

Figure 1 visualizes the minimum without and with the skip connection. This result reveals the effect of skip connection on the connectedness of the set of minima, which may lead to a new explanation of the effectiveness of ResNets (He et al., 2016) and DenseNets (Huang et al., 2017). We leave the connection between the topology of the minimum and the optimization and generalization properties of neural networks to future work.

# 5. Mode Connectivity

The previous section counts the connected components of the minimum and shows that the connectedness of the minimum is related to the symmetry of the loss function under certain conditions. In this section, we use this insight to explain recent empirical observations that with high probability two points in the minimum are connected, i.e. there is a large connected component. Proofs of this section appears in Appendix D.

Mode connectivity refers to the phenomenon that there exist high accuracy or low loss paths between two minima found by stochastic gradient descent (Garipov et al., 2018). Linear mode connectivity occurs when all points on the linear interpolation between two minima have low loss values. More recently, permutation of neurons is usually performed to align the two minima before evaluating linear mode connectivity (Entezari et al., 2022; Ainsworth et al., 2023). We use the term mode connectivity when we consider arbitrary curves and will specify linear mode connectivity when only linear interpolation is considered.

## 5.1. Mode connectivity up to permutation

For the family of linear neural networks defined in Section 4.1, we show that permutations allow us to connect points in the minimum that are not connected without permutation. Our results support the empirical observation that neuron alignment by permutation improves mode connectivity (Tatro et al., 2020).

Consider again the linear network (1) with invertible weights. When $l = 2$, the minimum of $L$ has two connected components corresponding to the two connected components of the $GL$ group. Any $g \in GL$ that is not on the identity component can take a point on one connected component of the minimum to the other.

**Lemma 5.1.** *Consider two points $(W_1, W_2), (W_1', W_2') \in L^{-1}(0)$ that are not connected in $L^{-1}(0)$. For any $g \in GL(h)$ such that $\det(g) < 0$, $g \cdot (W_1, W_2)$ and $(W_1', W_2')$*

*are connected in $L^{-1}(0)$.*

When the hidden dimension $h \geq 2$, there exists a permutation $g$ such that $\det(g) > 0$, and a permutation $g$ such that $\det(g) < 0$. Therefore, Lemma 5.1 implies the following result that all points on the minimum of $L$ are connected up to permutation.

**Proposition 5.2.** *Assume that $h \geq 2$. For all $(W_1, ..., W_l), (W_1', ..., W_l') \in L^{-1}(0)$, there exists a list of permutation matrices $P_1, ..., P_{l-1}$ such that $(W_1 P_1, P_1^{-1} W_2 P_2, ..., P_{l-2} W_{l-1} P_{l-1}, P_{l-1} W_l)$ and $(W_1', ..., W_l')$ are connected in $L^{-1}(0)$.*

The results above are examples where a larger part of the minimum becomes connected after a permutation. More generally, permutation improves mode connectivity in cases where an orbit is not connected due to the symmetry group comprising multiple connected components, the orbit does not reside on the same connected component of the minimum, and there exists a permutation that takes a point on one connected component of the group to another.

## 5.2. Failure case of linear mode connectivity

As an application of obtaining new minima from old ones using symmetries, we show that linear mode connectivity fails to hold in multi-layer regressions. The following proposition says that in neural networks with a homogeneous activation (such as leaky ReLU) between the last two layers, the error barrier in the linear interpolation between two solutions can be arbitrarily large.

**Proposition 5.3.** *Consider a loss function of the following form*

$$L : \quad Param \to \mathbb{R}, W = (W_1, ..., W_l) \mapsto$$
$$\|Y - W_l \sigma(W_{l-1} f(W_{l-2}, W_{l-3}, ..., W_1, X))\|_2^2, \quad (3)$$

*where $f$ is a function of $W_{l-2}, W_{l-3}, ..., W_1, X$, and $\sigma(cz) = c^k \sigma(z)$ for all $c \in \mathbb{R}$ and some $k > 0$. Assume that $\|Y\|_2 \neq 0$ and $L^{-1}(0) \neq \emptyset$. Also assume that $l \geq 2$. For any positive number $b > 0$, there exist $W, W' \in L^{-1}(0)$ that belong to the same connected component of $L^{-1}(0)$ and $0 < \alpha < 1$, such that $L((1-\alpha)W + \alpha W') > b$.*

The proof constructs a new point on the minimum from an existing one using the rescaling symmetry of homogeneous functions. The two points can be far apart since the orbit of this group action is unbounded. To provide intuition, Figure 2 visualizes the two points on the minimum of a two-layer network with weights of dimension $1 \times 1$ and the linear interpolation between them. The linear network used is a special case of a homogeneous network. Note that our result here does not contradict with the layer-wise connectivity result in (Adilova et al., 2024), as more than one layer of the two minima are different.

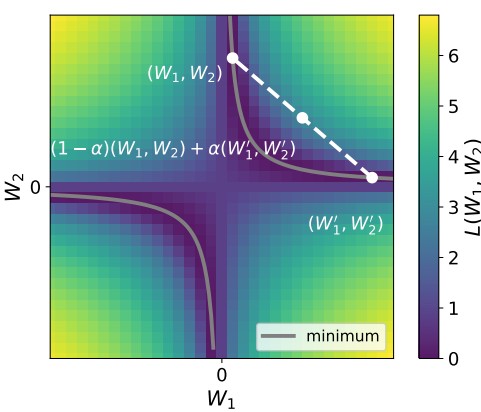

Figure 2: Interpolation between 2 minima of loss function $L(W_1, W_2) = ||Y - W_2 W_1 X||_2$ with 1 dimensional weights. Loss on the interpolation can be unbounded.

The loss function considered in Proposition 5.3 is significantly more general than those in Section 5.1. For the architecture, we only require the presence of a rescaling symmetry in the last two layers, and $f$ can be any neural network with any activation. Other assumptions of the proposition are also not excessively restrictive, as the labels $Y$ are rarely all zero, and there usually exists a minimum in common machine learning tasks.

Proposition 5.3 extends to cases where we allow certain permutations. The following proposition states that under additional assumptions, the error barrier in the linear interpolation is unbounded even with neuron permutations. The proof construction is similar to that of Proposition 5.3.

Let $S_n$ be the set of $n \times n$ permutation matrices, where $n$ is the number of columns of $W_l$.

**Proposition 5.4.** *Consider the loss function with the same set of assumptions in Proposition 5.3. Assume additionally that there does not exist a permutation $P$ such that every column of $P\sigma(W_{l-1}f(W_{l-2}, W_{l-3}, ..., W_1, X))$ is in the null space of $W_l$. For any positive number $b > 0$, there exist $(W_1, ..., W_l), (W_1', ..., W_l') \in L^{-1}(0)$ and $0 < \alpha < 1$, such that $(W_1, ..., W_{l-2}) = (W_1', ..., W_{l-2}')$ and*

$$\min_{P \in S_n} L\big((1-\alpha)(W_1, ..., W_l)$$
$$+ \alpha(W_1, ..., W_{l-2}, P^{-1}W_{l-1}, W_l P)\big) > b.$$

By including permutation, the setting in Proposition 5.4 is closer to the setting in which linear mode connectivity is empirically observed. However, the permutation in Proposition 5.4 is restricted to the first two layers, which does not rule out the possibility of lowering the loss barrier by including permutations of other neurons.

The proofs of Proposition 5.3 and 5.4 depend on the rescaling symmetry of homogenenous activation functions. For

other activations with known symmetries, similar results may be derived as using the large set of minimum obtained from the group action. Whether the loss barrier on the linear interpolation is bounded can depend on the compactness of the symmetry group and the curvature of the minimum. We leave a systematic investigation of the condition for linear mode connectivity to future work.

One possible reason why linear mode connectivity is observed in practice despite Proposition 5.4 is that only a small part of the minima is reachable by stochastic gradient descent due to implicit bias (Min et al., 2021), as other optimizers have been observed to find less connected minima (Altintas et al., 2023).

## 5.3. Linear mode connectivity of orbits

Symmetry accounts for a large part of the set of minima. In particular, given a known minimum $x$, the orbit of $x$ defines a set of points that are also minima. Although not all minima are on the same orbit of known symmetries, each orbit often contains a nontrivial set of minima. In this section, we examine the error barrier of linear interpolations of minima restricted to an orbit of parameter symmetries.

When the architecture contains a multiplication of two weight matrices $W_2 W_1$, where $W_2 \in \mathbb{R}^{m \times h}, W_1 \in \mathbb{R}^{h \times n}$, there is a $GL_h$ symmetry that acts on $(W_1, W_2)$ by $g \cdot (W_1, W_2) = (gW_1, W_2 g^{-1})$ for $g \in GL_h$. The following proposition states that a point on the linear interpolation of two points in the same orbit can be far away from the orbit.

**Proposition 5.5.** *Let $A \in \mathbb{R}^{n \times n}$ be an invertible matrix. Let set $S = \{(W_1, W_2) : W_1, W_2 \in \mathbb{R}^{n \times n}, W_1 W_2 = A\}$. For any positive number $b > 0$, there exist $W', W'' \in S$ and $0 < \alpha < 1$, such that $\min_{\hat{W} \in S} \| ((1-\alpha)W' + \alpha W'') - \hat{W}\|_2 > b$.*

The structure in the form of $W_1 W_2$ is not uncommon in deep learning architectures. Notably, the parameter matrices for queries and keys in the attention function are multiplied directly in this manner (Vaswani et al., 2017), thus admitting the $GL_h$ symmetry and having orbits with properties given by Proposition 5.5.

While the error barrier in the linear interpolation of two minima can be unbounded (Proposition 5.3), this typically occurs when the parameters are allowed to be arbitrarily large. Constraining the parameters to remain bounded ensures that the loss barrier is bounded above. The following proposition makes this intuition precise for the set of minima consisting of a particular orbit.

**Proposition 5.6.** *Consider the loss function with the same set of assumptions in Proposition 5.3. Let $W \in L^{-1}(0)$ be a point on the minimum. Consider the multiplicative group of positive real numbers $\mathbb{R}^+$ that acts on $L^{-1}(0)$ by $g \cdot (W_1, ..., W_l) = (W_1, ..., W_{l-2}, gW_{l-1}, W_l g^{-k})$, where*

$g \in \mathbb{R}^+$. *Then there exists a positive number $b > 0$, such that for all $0 < \alpha < 1$ and $W' \in Orbit(W)$ with $\|W_i'\|_2 < c$ for all $i$ and some $c > 0$, the loss value for points on the linear interpolation $L\left((1-\alpha)W + \alpha W'\right) < b$.*

Proposition 5.5 and 5.6 are two examples where the knowledge of parameter symmetry enables analysis of the linear connectivity of subsets of minima. As more continuous symmetries are characterized (e.g. the nonlinear symmetries in Zhao et al. (2023)), these analysis can potentially be extended to even larger parts of the set of minima.

# 6. Curves on Minimum from Group Actions

The paths connecting two points in the set of minima may not be linear. Previously, these paths were discovered empirically by finding parametric curves on which the expected loss is minimized (Garipov et al., 2018). Using parameter space symmetry, we uncover an alternative and principled way to find curves on the minimum.

## 6.1. Symmetry induced curves

Suppose the loss function $L :$ Param $\to \mathbb{R}$ is invariant with respect to some Lie group $G$. Consider the following curve for a point $\boldsymbol{w} \in$ Param and $M \in \mathrm{Lie}(G)$:

$$\gamma_M : \mathbb{R} \times \text{Param} \to \text{Param},$$
$$\gamma_M(t, \boldsymbol{w}) = \exp\left(tM\right) \cdot \boldsymbol{w}. \tag{4}$$

Since $\exp\left(tM\right) \in G$ and the action of $G$ preserves the value of $L$, every point on $\gamma_M$ is in the same $L$ level set as $\boldsymbol{w}$. This provides a way to find a curve of constant loss between two points that are in the same orbit. Concretely, given two points $\boldsymbol{w}_1$ and $\boldsymbol{w}_2 = g \cdot \boldsymbol{w}_1$, let $\gamma$ be the following curve:

$$\gamma : [0, 1] \times G \times \text{Param} \to \text{Param},$$
$$\gamma(t, g, \boldsymbol{w}) = \exp\left(t \log(g)\right) \cdot \boldsymbol{w}. \tag{5}$$

Note that $\gamma(0, g, \boldsymbol{w}_1) = \boldsymbol{w}_1$, $\gamma(1, g, \boldsymbol{w}_1) = \boldsymbol{w}_2$, and $L(\gamma(t, g, \boldsymbol{w}_1)) = L(\boldsymbol{w}_1) = L(\boldsymbol{w}_2)$ for all $t \in [0, 1]$. Hence, $\gamma$ is a curve that connects the points $\boldsymbol{w}_1$ and $\boldsymbol{w}_2$, and every point on $\gamma$ has the same loss value as $L(\boldsymbol{w}_1) = L(\boldsymbol{w}_2)$.

For a group $G$, the curve $\gamma$ is defined when the map $\cdot :$ $G \times$ Param $\to$ Param is continuous and id $\cdot \boldsymbol{w} = \boldsymbol{w}$ for all $\boldsymbol{w} \in$ Param, even if it is not a group action or does not preserve loss. However, when $\cdot$ does not preserve loss, the loss can change on $\gamma$. Consider our two-layer network and the following map:

$$\cdot : GL(h, \mathbb{R}) \times \text{Param} \to \text{Param}$$
$$g \cdot (U, V) = (U\sigma(VX)\sigma(gVX)^\dagger, gV). \tag{6}$$

When $\sigma$ is the identity function, $\cdot$ preserves the loss value, and $\gamma$ defines a curve on the minimum. In general, the map

(6) does not preserve loss when batch size $k$ is larger than hidden dimension $h$. However, the maximum change of loss on $\gamma$ can be bounded as follows.

**Proposition 6.1.** *Let $(U, V) \in$ Param, and $(U', V') = g \cdot (U, V)$. Then*

$$\|U\sigma(VX) - U'\sigma(V'X)\| \leq \|U\sigma(VX)\|. \tag{7}$$

We demonstrate Proposition 6.1 empirically using a set of two-layer networks with various parameter space dimensions. Specifically, we construct networks in the form of $\|U\sigma(VX) - Y\|^2$, with $\sigma$ being the sigmoid function, $X \in \mathbb{R}^{n \times k}, Y \in \mathbb{R}^{m \times k}$, and $(U, V) \in$ Param $= \mathbb{R}^{m \times h} \times \mathbb{R}^{h \times n}$. We create 100 such networks, each with $m, h, n, k$ randomly sampled from integers between 2 and 100. In each network, elements in $X$ and $Y$ are sampled independently from a normal distribution, and $U, V$ are randomly initialized. After training with SGD, we compute $(U', V') = g \cdot (U, V)$ using (6) with a random invertible matrix $g$. We then plot $\|U\sigma(VX)\|$ against $\|U\sigma(VX) - U'\sigma(V'X)\|$ in Figure 3(a). All points are above the line $y = x$, as predicted by Proposition 6.1.

While the map (6) is not a group action in general, it connects more points in the set of minima than only using known symmetries, and the points on the connecting curves have bounded loss. Figure 3(b-c) shows that the loss on the curves induced by approximate symmetries remains relatively low, compared to the loss on the linear interpolation between the two ends of these curves. We consider a two layer network with loss function $\|W_2\sigma(W_1X) - Y\|$, with $\sigma$ being a leaky ReLU function, $X \in \mathbb{R}^{16 \times 8}, Y \in \mathbb{R}^{64 \times 8}$, and $(W_1, W_2) \in$ Param $= \mathbb{R}^{32 \times 16} \times \mathbb{R}^{32}$. In the figures, $\gamma$ denotes a curve obtained using Equation (5) together with (6). The starting point of $\gamma$ is a minimum found by SGD. Both $\gamma$ and the linear interpolation are parametrized by $t \in [0, 1]$. Compared to the linear interpolation between the two end points of $\gamma$, the loss on $\gamma$ is consistently lower. Figure 3(c) uses group elements with larger magnitudes, resulting in a larger distance between $\gamma(0)$ and $\gamma(1)$, which might explain the higher loss barrier on their linear interpolation.

## 6.2. Approximate linear connectivity under bounded curvature of minima

Knowing the explicit expression of connecting curves brings new insight into when linear mode connectivity approximately holds. In particular, these expressions provide information about the curvature of the curves. If the curvatures are small, then there exists an approximately straight line connecting any two minima along which the loss remains close to its minimum value.

Consider a loss level set $L^{-1}(c) = \{\boldsymbol{w} \in$ Param $: L(\boldsymbol{w}) = c\}$ with some $c \in \mathbb{R}$. Suppose we have two points

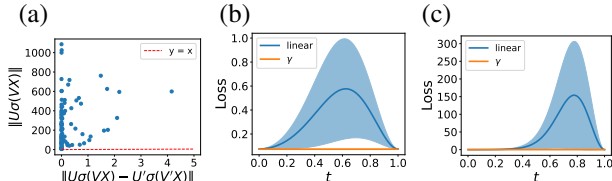

Figure 3: (a) Empirical validation of Proposition 6.1. (b-c) The loss on the curves induced by approximate symmetries ($\gamma$) remains relatively low, compared to the loss on the linear interpolation between the two ends of these curves. (b) and (c) differ by the magnitude of the group element used. The loss is averaged over 5 random curves.

$\boldsymbol{w}_1, \boldsymbol{w}_2 \in L^{-1}(c)$ connected by a smooth curve $\gamma$ lying entirely within $L^{-1}(c)$. The curvature of $\gamma$ can be written as $\kappa(\gamma, t) = \frac{\|T'(t)\|}{\|\gamma'(t)\|}$, where $\gamma' = \frac{d\gamma}{dt}$ and $T(t) = \frac{\gamma'(t)}{\|\gamma'(t)\|}$. If the curvature of this curve is small or bounded, we can show that there exists an approximately straight line connecting $\boldsymbol{w}_1$ and $\boldsymbol{w}_2$ that remains close to $L^{-1}(c)$. Additionally, if $L$ is Lipschitz continuous, its value remains close to $c$ along this line segment. We formalize this with the following theorem.

**Theorem 6.2.** *Let $L^{-1}(c) \subset$ Param, with $c \in \mathbb{R}$, be a level set of the loss function $L :$ Param $\to \mathbb{R}$. Let $\gamma : [0, 1] \to L^{-1}(c)$ be a smooth curve in $L^{-1}(c)$ connecting two points $\boldsymbol{w}_1 = \gamma(0)$ and $\boldsymbol{w}_2 = \gamma(1)$. Suppose the curvature $\kappa(t)$ of $\gamma$ satisfies $\kappa(t) \leq \kappa_{\max}$ for all $t \in [0, 1]$.*

*Let $S$ be the straight line segment connecting $\boldsymbol{w}_1$ and $\boldsymbol{w}_2$. Then, for any point $\boldsymbol{w}$ on $S$, the distance to $L^{-1}(c)$ is bounded by*

$$\text{dist}(\boldsymbol{w}, L^{-1}(c)) \leq d_{\max}, \tag{8}$$

*with*

$$d_{\max} = \frac{1}{\kappa_{\max}} \left( 1 - \sqrt{1 - \left( \frac{\kappa_{\max} \|\boldsymbol{w}_2 - \boldsymbol{w}_1\|_2}{2} \right)^2} \right).$$

*Furthermore, assuming $L$ is Lipschitz continuous with Lipschitz constant $C_L$, the loss at any point $\boldsymbol{w}$ on $S$ satisfies*

$$|L(\boldsymbol{w}) - c| \leq C_L d_{\max}. \tag{9}$$

When the group action induces curves with bounded curvature, Theorem 6.2 applies. Since the minimum is also a level set of $L$, Theorem 6.2 provides a sufficient condition for linear mode connectivity to approximately hold. When the curvature of the minimum is small, points on the minimum are approximately connected through nearly straight paths along with the loss does not increase significantly. If $\kappa_{\max} \|\boldsymbol{w}_2 - \boldsymbol{w}_1\|$ is small, we can use the first-order approximation of the square root and obtain $d_{\max} \approx \frac{\kappa_{\max} \|\boldsymbol{w}_2 - \boldsymbol{w}_1\|_2^2}{8}$.

# 7. Discussion

In this work, we study topological properties of the loss level sets by relating their topology to the topology of symmetry groups. Specifically, we derive the number of connected components of full-rank multi-layer networks with and without skip connections, and prove mode connectivity up to permutation for full-rank linear regressions. Using symmetry in the parameter space, we construct an explicit expression for curves that connect two points in the same orbit. The explicit expressions allow us to obtain the curvature of these curves, which are useful to bound the loss barrier on linear interpolation between minima.

For practitioners, these results motivate concrete strategies—and cautions—for tasks that navigate the loss landscape, including model merging, ensembling, and fine-tuning:

- One can build low-loss curves explicitly using known parameter symmetries. This gives a principled and efficient way to obtain new minima from old ones, potentially useful for (1) generating model ensembles with low cost; (2) improving model alignment by allowing a much larger group of transformations than permutation; and (3) mitigating catastrophic forgetting in fine-tuning by constraining updates to the symmetry-induced manifold of the pretraining minimum.

- The connectedness of minima supports the practice of model merging and ensembling, even when models are trained separately. In addition to permutation, many other symmetry transformations can connect solutions that would otherwise appear very different.

- Linear interpolation between minima is not guaranteed to lead to better models, despite its widespread use. This highlights the need to evaluate whether the minima found by specific learning algorithms are approximately connected before averaging models directly.

Extending these results to nonlinear networks is a challenging yet exciting future direction. A full characterization of the minima in non-linear settings requires identifying the complete set of symmetries—an open problem for many architectures—and analyzing how the resulting orbits intersect, which becomes increasingly complex as the number of orbits grows. One approach is through approximate symmetries, such as those in Section 6. Another is by continuously deforming the network or its minimum and studying its behavior in the limit as the network approaches a linear regime. Additionally, many modern architectures retain large continuous symmetry groups, particularly in components like self attention or normalization layers. As we saw in Section 5.3, even partial knowledge of symmetries in a network can yield valuable structural information about its minima.

# Acknowledgements

We thank Iordan Ganev for helpful comments on proofs in Appendix A. This work was supported in part by the U.S. Army Research Office under Army-ECASE award W911NF-07-R-0003-03, the U.S. Department Of Energy, Office of Science, IARPA HAYSTAC Program, and NSF Grants #2205093, #2146343, #2134274, #2107256, #2134178, CDC-RFA-FT-23-0069, DARPA AIE FoundSci and DARPA YFA.

# Impact Statement

This work advances the theoretical understanding of mode connectivity in neural network loss landscapes through parameter space symmetries, with potential applications in model merging and fine-tuning. We do not identify specific ethical concerns but recognize that implications may vary depending on the domain of application.

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

# Appendix

## A. Background

This section contains additional background in general topology and proofs for statements in Section 3. We refer readers to (Lee, 2010) for a more detailed introduction to this topic.

### A.1. Connected components

Consider two topological spaces $X$ and $Y$. A map $f : X \to Y$ is *continuous* if for every open subset $U \subseteq Y$, its preimage $f^{-1}(U)$ is open in $X$. If $X$ and $Y$ are metric spaces with metrics $d_X$ and $d_Y$ respectively, this is equivalent to the delta-epsilon definition. That is, $f$ is continuous if at every $x \in X$, for any $\epsilon > 0$ there exists $\delta > 0$ such that $d_X(x, y) < \delta$ implies $d_Y(f(x), f(y)) < \epsilon$ for all $y \in X$.

A topological space is *connected* if it cannot be expressed as the union of two disjoint, nonempty, open subsets. A topological space $X$ is *path connected* if for every $p, q \in X$, there is a continuous map $f : [0, 1] \to X$ such that $f(0) = p$ and $f(1) = q$. Path connectedness implies connectedness, but the converse is not true (Lee, 2010). (Nguyen, 2019) studies the path connectedness of sublevel sets of loss functions.

The following theorem is the main intuition of this paper and will appear frequently in proofs.

**Theorem A.1** (Theorem 4.7 in (Lee, 2010), Theorem 3.1 in the Main Text). *Let $X, Y$ be topological spaces and let $f : X \to Y$ be a continuous map. If $X$ is connected, then $f(X)$ is connected.*

A map $f$ is a *homeomorphism* from $X$ to $Y$ if $f$ is bijective and both $f$ and $f^{-1}$ are continuous. $X$ and $Y$ are *homeomorphic* if such a map exists. A *(connected) component* of a topological space $X$ is a maximal nonempty connected subset of $X$. The components of $X$ form a partition of $X$. The next two corollaries of Theorem A.1 show that connectedness and the number of connected components are topological properties. That is, they are preserved under homeomorphisms.

**Corollary A.2.** *Let $f : X \to Y$ be a homeomorphism from $X$ to $Y$, and let $U \subseteq X$ be a subset of $X$ with the subspace topology. Then $U$ is connected if and only if $f(U) \subseteq Y$ is connected.*

*Proof.* By the definition of homeomorphism, $f$ and $f^{-1}$ are continuous. From Theorem A.1, if $U \in X$ is connected, then $f(U) \in Y$ is connected. Similarly, if $f(U)$ is connected, then $f^{-1}(f(U)) = U$ is connected. $\square$

**Corollary A.3.** *Let $X$ be a topological space that has $N$ components. Let $Y$ be a topological space homeomorphic to $X$. Then $Y$ has $N$ components.*

*Proof.* Let $C_1, ..., C_N$ be the components of $X$. Let $f$ be a homeomorphism from $X$ to $Y$. Since $f$ is bijective and $C_1, ..., C_N$ is a partition of $X$, $f(C_1), ..., f(C_N)$ is a partition of $Y$. From Theorem A.1, since $C_1, ..., C_N$ are all connected, so are $f(C_1), ..., f(C_N)$.

Lastly, we need to show that $f(C_1), ..., f(C_N)$ are maximally connected. Suppose there exists a set $U \subseteq Y$, such that $U \not\subseteq f(C_i)$ and $f(C_i) \cup U$ is connected for some $i$. Then by Theorem A.1, $f^{-1}(f(C_i) \cup U) \supset C_i$ is connected in $X$. This contradicts the fact that $C_i$ is a maximal component in $X$. Therefore, $f(C_1), ..., f(C_N)$ are maximally connected.

Since $f(C_1), ..., f(C_N)$ partitions $Y$ and are maximally connected, $Y$ has $N$ components. $\square$

Another consequence of Theorem A.1 is the following upper bound on the number of components of the image of a continuous map.

**Proposition A.4.** *Let $f : X \to Y$ be a continuous map. The number of components of the image $f(X) \subseteq Y$ is at most the number of components of $X$.*

*Proof.* Let $C_1, ..., C_N$ be the components of $X$. Since $C_i$ is continuous and the action is continuous, according to Theorem A.1, $f(C_i)$ is continuous for all $i \in \{1, ..., N\}$. Additionally, since $\bigcup_{i=1}^{N} C_i = X$, we have $\bigcup_{i=1}^{N} f(C_i) = f(X)$. Therefore, there is a surjective map from $\{f(C_1), ..., f(C_N)\}$ to the set of components of $f(X)$, which implies that $f(X)$ has at most $N$ components. $\square$

Let $X_1, ..., X_n$ be topological spaces. The *product space* is their Cartesian product $X_1 \times ... \times X_n$ endowed with the product topology. Denote $\pi_0(X)$ as the set of connected components of a space $X$. The following proposition provides a way to count the components of a product space.

**Proposition A.5.** *Consider $n$ topological spaces $X_1, ..., X_n$. Then $|\pi_0(X_1 \times ... \times X_n)| = \prod_{i=0}^{n} |\pi_0(X_i)|$.*

*Proof.* When $n = 1$, the number of components of the product space is $|\pi_0(X_1)|$.

For the $n > 1$ case, since $X_1 \times ... \times X_n = (X_1 \times ... \times X_{n-1}) \times X_n$, it suffices to show that $|\pi_0(A \times B)| = |\pi_0(A)||\pi_0(B)|$ for any topological spaces $A$ and $B$. Let $f : \pi_0(A) \times \pi_0(B) \to \pi_0(A \times B)$ be the map that assigns $C \in \pi_0(A) \times \pi_0(B)$ to the element in $\pi_0(A \times B)$ that contains $C$. Then $f$ is surjective because $\pi_0(A) \times \pi_0(B)$ forms a partition of $A \times B$. To prove that $f$ is injective, suppose that $f(C_1) = f(C_2)$ for $C_1, C_2 \in \pi_0(A) \times \pi_0(B)$. Consider the projection $\pi_A : A \times B \to A$. Since $\pi_A$ is continuous and $C_1, C_2$ belong to the same component of $A \times B$, $\pi_A(C_1)$ and $\pi_A(C_2)$ belong to the same component of $A$. Similarly, $\pi_B(C_1)$ and $\pi_B(C_2)$ belong to the same component of $B$ under the projection $\pi_B : A \times B \to B$. Since all components of $A$ and $B$ are maximally connected, we have $C_1 = C_2$, which implies that $f$ is injective. Since $f$ is a bijection from $\pi_0(A) \times \pi_0(B)$ to $\pi_0(A \times B)$, $|\pi_0(A \times B)| = |\pi_0(A)||\pi_0(B)|$. $\qquad\square$

## A.2. Groups

A *group* is a set $G$ together with a composition law, written as juxtaposition, that satisfies associativity, $(ab)c = a(bc)$ $\forall\, a, b, c \in G$, has an identity $1$ such that $1a = a1 = a \,\forall\, a \in G$, and for all $a \in G$, there exists an inverse $b$ such that $ab = ba = 1$. An *action* of a group $G$ on a set $S$ is a map $\cdot : G \times S \to S$ that satisfies $1 \cdot s = s$ for all $s \in S$ and $(gg') \cdot s = g \cdot (g' \cdot s)$ for all $g, g'$ in $G$ and all $s$ in $S$. The *orbit* of $s \in S$ is the set $O(s) = \{s' \in S \mid s' = gs \text{ for some } g \in G\}$.

A *topological group* is a group $G$ endowed with a topology such that multiplication and inverse are both continuous. A recurring example is the general linear group $GL_n(\mathbb{R})$, with the subspace topology obtained from $\mathbb{R}^{n^2}$. The group $GL_n(\mathbb{R})$ has two connected components, which correspond to matrices with positive and negative determinant.

The *product* of groups $G_1, ..., G_n$ is a group denoted by $G_1 \times ... \times G_n$. The set underlying $G_1 \times ... \times G_n$ is the Cartesian product of $G_1, ..., G_n$. The group structure is defined by identity $(1, ..., 1)$, inverse $(g_1, ..., g_n)^{-1} = (g_1^{-1}, ..., g_n^{-1})$, and multiplication rule $(g_1, ..., g_n)(g_1', ..., g_n') = (g_1 g_1', ..., g_n g_n')$.

## A.3. Relating connectedness of groups, orbits, and level sets

From Theorem 3.1, continuous maps preserve connectedness. Through continuous actions, we study the connectedness of orbits and level sets by relating them to the connectedness of more familiar objects such as the general linear group. Establishing a homeomorphism from the group to the set of minima requires the symmetry group's action to be continuous, transitive, and free. Here we only assume the action to be continuous and try to bound the number of components of the orbits.

As an immediate consequence of Proposition A.4, an orbit cannot have more components than the group.

**Corollary A.6.** *Assume that the action of a group $G$ on $S$ is continuous. Then the number of connected components of orbit $O(s)$ is smaller than or equal to the number of connected components of $G$, for all $s$ in $S$.*

*Proof.* An orbit $O(s)$ is the image of the group action, which we assume to be continuous. The result follows from Proposition A.4. $\qquad\square$

Let $X$ be a topological space and $L : X \to \mathbb{R}$ a continuous function on $X$. A topological group $G$ is said to be a *symmetry group* of $L$ if $L(g \cdot x) = L(x)$ for all $g \in G$ and $x \in X$. In this case, the action can be defined on a level set of $L$, $L^{-1}(c)$ with a $c \in \mathbb{R}$, as $G \times L^{-1}(c) \to L^{-1}(c)$. If the minimum of $L$ consists of a single orbit, Corollary A.6 extends immediately to the number of components of the minimum.

**Corollary A.7.** *Let $L$ be a function with a symmetry group $G$. If the minimum of $L$ consists of a single $G$-orbit, then the number of connected components of the minimum is smaller or equal to the number of connected components of $G$.*

Generally, symmetry groups do not act transitively on a level set $L^{-1}(c) \in X$. In this case, the connectedness of the orbits does not directly inform the connectedness of the level set.

**Proposition A.8.**

1. *There exists a space $X$ and a group $G$ with an action on $X$, such that each orbit for the group action is connected and $X$ is not connected.*

2. *There exists a space $X$ and a group $G$ with an action on $X$, such that each orbit for the group action is disconnected and $X$ is connected.*

*Proof.* For part (a), consider a subspace of $\mathbb{R}^2$, $X = X_1 \cup X_2$ where $X_1 = \{(x, y) : x = 0, y > 0\}$ and $X_2 = \{(x, y) : x = 1, y > 0\}$. The space $X$ is not connected. Let $G$ be the multiplicative group of positive real numbers and act on $X$ by multiplication on the second coordinate. Then there are two orbits, $X_1$ and $X_2$, which are both connected.

For part (b), consider the space $X = \mathbb{R}^2 \setminus \{0\}$. Then $X$ is connected. Let $G$ be the multiplicative group of real numbers, which acts on $X$ by multiplication on both coordinates. That is, $g \cdot (x_1, x_2) = (gx, gx_2), \forall (x_1, x_2) \in X, \forall g \in G$. The orbit of any point $(x_1, x_2) \in X$ is not connected. $\qquad\square$

Nevertheless, since the set of orbits partitions the space, we can use the following bound on the number of components of the space.

**Proposition A.9.** *Let $X$ be a topological space and let $X = \coprod_i X_i$ be a partition of $X$ into disjoint subspaces. Then $|\pi_0(X)| \leq \sum_i |\pi_0(X_i)|$.*

*Proof.* Let $S = \{A \subseteq X : \exists i, A \text{ is a component of } X_i\}$ be the union of the components of the subspaces. Then $S$ is a partition of $X$, and every element in $S$ is connected. Therefore, there is a surjective map from $S$ to $\pi_0(X)$, defined by mapping each $s \in S$ to the element of $\pi_0(X)$ that includes $s$. This implies that $|\pi_0(X)| \leq |S| = \sum_{i=1}^n |\pi_0(X_i)|$. $\qquad\square$

Consider a topological space $X$ and a group $G$ that acts on $X$. Let $O = \{O_1, ..., O_n\}$ be the set of orbits. By Proposition A.9, the number of components of the orbits give the following upper bound on the number of components of the space: $|\pi_0(X)| \leq \sum_{i=1}^n |\pi_0(O_i)|$.

## B. Additional Related Work

**Topological approaches in machine learning.**    Topology has been applied in other areas of machine learning, particularly through tools such as persistent homology, to study the structure of data manifolds and training dynamics (Chazal & Michel, 2021). For example, prior work has used topological data analysis (TDA) to study the shape of activation patterns, understand generalization, and visualize learning trajectories (Gabrielsson & Carlsson, 2019).

**Bounds on the number of connected components.**    For neural networks that are systems of polynomials, the number of critical points can be upper bounded using methods in algebraic geometry. Mehta et al. (2021) shows that after adding a generalized L2 regularization, there are no positive-dimensional solutions in deep linear networks with mean squared error. They observe that the critical points, which satisfy $\nabla L = 0$, form the solution set of a system of polynomial equations. They then provide two upper bounds, the classical Bezout bound (CBB) and the Bernshtein-Kushnirenko-Khovanskii Bound (BKK), on the number of isolated complex critical points. Bharadwaj & Hoşten (2023) improves the upper bound for neural networks with one hidden layer and one training data point. Kohn et al. (2022) provide bounds on the number of critical points in the function space for linear convolutional networks. Since proving the exact number of connected components of a minimum is not always easy, a possible future direction is to derive Bezout and BKK bounds on the number of connected components for various architectures and perhaps extend this analysis beyond polynomial neural networks.

## C. Proofs in Section 4

**Proposition 4.1.** *There is a homeomorphism between $L^{-1}(0)$ and $(\mathrm{GL}_h)^{l-1}$.*

*Proof.* Recall that $W_1, ..., W_n, X, Y$ are matrices in $\mathbb{R}^{h \times h}$, and $X, Y$ are both full rank. Consider the map

$$f : (\mathrm{GL}_h)^{l-1} \to L^{-1}(0), \quad (g_1, ..., g_{l-1}) \mapsto (g_1 X^{-1}, g_2, ..., g_{l-1}, Y \prod_i^{l-1} g_i^{-1}). \tag{10}$$

The inverse $f^{-1} : (W_1, ..., W_l) \mapsto (W_1 X, W_2, W_3, ..., W_{l-1})$ is well defined, because $X, W_1, W_2, W_3, ..., W_{l-1}$ are all full-rank. Since both $f$ and $f^{-1}$ are continuous, $f$ is a homeomorphism between $(\mathrm{GL}_h)^{l-1}$ and $L^{-1}(0)$. $\quad\square$

**Corollary 4.2.** *The minimum of $L$ has $2^{l-1}$ connected components.*

*Proof.* From Proposition 4.1, $L^{-1}(0)$ is homeomorphic to $(\mathrm{GL}_h)^{l-1}$. According to Corollary A.3, this implies that $L^{-1}(0)$ has the same number of connected components as $(\mathrm{GL}_h)^{l-1}$. From Proposition A.5, $GL_h(\mathbb{R})^{l-1}$ has $2^{l-1}$ connected components. Therefore, $L^{-1}(0)$ has $2^{l-1}$ connected components. $\quad\square$

**Proposition 4.3.** *Let $n = 1$. Assume that $X, Y \neq 0$. When $\varepsilon = 0$, the minimum of $L$ has 4 connected components. When $\varepsilon \neq 0$, the minimum of $L$ has 3 connected components.*

*Proof.* When $\varepsilon = 0$, the skip connection is effectively removed, and the loss function (2) reduces to (1). By Corollary 4.2, the minimum of $L$ has 4 connected components. In the rest of the proof, we consider the case where $\varepsilon \neq 0$.

Let $(W_{1_0}, W_{2_0}, W_{3_0}) = (I, (\alpha - \varepsilon)I, \alpha^{-1} Y X^{-1})$, where $\alpha \in \mathbb{R}$ is an arbitrary number such that $\alpha \neq \varepsilon$ and $\alpha \neq 0$. Then $(W_{1_0}, W_{2_0}, W_{3_0})$ is a point in $L^{-1}(0)$. Define set $G_1 = \{g \in R^{h \times h} : \det(g W_{2_0} W_{1_0} X + \varepsilon X) \neq 0\}$. Let $a : GL_1 \times G_1 \to$ Param be the following map:

$$\begin{aligned} g_1, g_2 \mapsto (&g_1 W_{1_0}, \\ &g_2 W_{2_0} g_1^{-1}, \\ &W_{3_0}(W_{2_0} W_{1_0} X + \varepsilon X)(g_2 W_{2_0} W_{1_0} X + \varepsilon X)^{-1}). \end{aligned} \tag{11}$$

From the definition of $G_1$, $(g_2 W_{2_0} W_{1_0} X + \varepsilon X)$ is invertible, so $a$ is well defined. Additionally, we have $L(a(g_1, g_2)) = L(W_{1_0}, W_{2_0}, W_{3_0}) = 0, \forall g_1, g_2 \in GL_1 \times G_1$. Therefore, denoting the image of $a$ as $S_1$, we have $S_1 \subseteq L^{-1}(0)$.

Let $S_0 = \{(W_1, W_2, W_3) : W_3 = Y(\varepsilon X)^{-1}$ and $W_1 = 0\}$ if $\varepsilon \neq 0$, or $\emptyset$ otherwise. For $(W_1, W_2, W_3) \in S_0$, we have $L(W_1, W_2, W_3) = ||Y - Y(\varepsilon X)^{-1}(0 + \varepsilon X)||_2 = 0$. Therefore, $S_0 \subseteq L^{-1}(0)$.

We then show that the minimum of $L$ is the union of $S_1$ and $S_0$. Consider a point $(W_1, W_2, W_3) \in L^{-1}(0)$. If $W_1 = 0$, then $\varepsilon \neq 0$, otherwise $(W_1, W_2, W_3)$ cannot be in $L^{-1}(0)$. In this case, $W_3$ must equal to $Y(\varepsilon X)^{-1}$, and $(W_1, W_2, W_3) \in S_0$. If $W_1 \neq 0$, then $W_1 W_{1_0}^{-1} \in GL_1$ and $W_2 W_1 W_{1_0}^{-1} W_{2_0}^{-1} \in G_1$. The second part is due to $W_2 W_1 W_{1_0}^{-1} W_{2_0}^{-1} W_{2_0} W_{1_0} X + \varepsilon X = W_2 W_1 X + \varepsilon X \neq 0$ since $(W_1, W_2, W_3) \in L^{-1}(0)$. In this case we have $(W_1, W_2, W_3) = a(W_1 W_{1_0}^{-1}, W_2 W_1 W_{1_0}^{-1} W_{2_0}^{-1})$, which means that $(W_1, W_2, W_3) \in S_1$.

The number of connected components of $S_1$ and $S_0$ can be obtained from their structures. Since $W_{2_0} W_{1_0} X \neq 0$, there is a homeomorphism between $G_1$ and $GL_1$ defined by the map

$$f : G_1 \to GL_1, g \mapsto g W_{2_0} W_{1_0} X + \varepsilon X \tag{12}$$

with inverse $f^{-1} : GL_1 \to G_1, g \mapsto \varepsilon(g - \varepsilon X)(W_{2_0} W_{1_0} X)^{-1}$. Since $a$ is also a homeomorphism, its image $S_1$ is homeomorphic to $GL_1 \times GL_1$ and has 4 connected components. When $\varepsilon \neq 0$, $S_0$ is a line and thus has 1 connected component.

The last part of the proof shows the connectedness of the connected components of $S_1$ and $S_0$. Let $G_1^+ = \{g_2 \in G_1 : f(g_2) \in GL^{sign(\varepsilon X)}\}$ be the connected component in $G_1$ that correspond to $GL^{sign(\varepsilon X)}$, and $G_1^- = \{g_2 \in G_1 : f(g_2) \in GL^{-sign(\varepsilon X)}\}$ be the component that correspond to $GL^{-sign(\varepsilon X)}$. For convenience, we name the connected components of $Im(a)$ as follows:

$$\begin{aligned} C_1 &= \{(W_1, W_2, W_3) \in \text{ Param} : (W_1, W_2, W_3) = a(g_1, g_2), g_1 \in GL^+, g_2 \in G_1^+\} \\ C_2 &= \{(W_1, W_2, W_3) \in \text{ Param} : (W_1, W_2, W_3) = a(g_1, g_2), g_1 \in GL^-, g_2 \in G_1^+\} \\ C_3 &= \{(W_1, W_2, W_3) \in \text{ Param} : (W_1, W_2, W_3) = a(g_1, g_2), g_1 \in GL^+, g_2 \in G_1^-\} \\ C_4 &= \{(W_1, W_2, W_3) \in \text{ Param} : (W_1, W_2, W_3) = a(g_1, g_2), g_1 \in GL^-, g_2 \in G_1^-\} \end{aligned}$$

Note that for $(W_1, W_2, W_3) \in S_1$, there exists a (unique) $g_2 \in G_1$ such that we can write $W_3$ as

$$W_3 = W_{3_0}[W_{2_0}W_{1_0}X + \varepsilon X][g_2 W_{2_0}W_{1_0}X + \varepsilon X]^{-1}) = Yf(g_2)^{-1}.$$

Following from the definition of $G_1^+$, for a point $(W_1, W_2, W_3)$ in $C_1$ or $C_2$, $sign(W_3) = sign(Y(\varepsilon X)^{-1})$. Additionally, when $g_2$ is close to 0, $g_2$ belongs to $G_1^+$. The boundary of both $C_1$ and $C_2$ contain a point in $S_0$:

$$\lim_{g_1 \to 0^+} a(g_1, g_1) = \lim_{g_1 \to 0^-} a(g_1, g_1) = (0, \alpha - \varepsilon, Y(\varepsilon X)^{-1}) \in S_0.$$

Therefore, both $C_1$ and $C_2$ are connected to $S_0$.

For points in $C_3$ and $C_4$, $sign(W_3) \neq sign(Y(\varepsilon X)^{-1})$. Therefore, no point in $C_3$ or $C_4$ can be sufficiently close to $S_0$. As a result, these components are not connected to $S_0$. In summary, when $\varepsilon \neq 0$, $S_0$ connects 2 components of $S_1$, and the minimum of $L$ has 3 connected components. $\qquad\square$

We note that connectedness alone does not imply easy connectivity in the sense of short or simple paths between solutions. Being in the same connected components is a necessary condition for connectivity, but a single component may still contain complex geometry necessitating complicated connecting paths.

Defining the ease of connectivity is subtle. One natural measure is the parametric complexity of the connecting curves, quantifiable by their degree if polynomial, or number of segments if piece-wise. Another possible definition for easy connectivity would be low curvature of the minimum manifold or short geodesic distance between two points on it. As we saw in Section 6.2, low curvature implies that linear interpolation stays near the manifold. Other potential definitions include whether the connecting curve has an analytical expression, or how many points are needed to approximate it within a certain error. It would be interesting to examine these properties for symmetry-induced connecting curves.

## D. Proofs in Section 5

**Lemma 5.1.** *Consider two points $(W_1, W_2), (W_1', W_2') \in L^{-1}(0)$ that are not connected in $L^{-1}(0)$. For any $g \in GL(h)$ such that $det(g) < 0$, $g \cdot (W_1, W_2)$ and $(W_1', W_2')$ are connected in $L^{-1}(0)$.*

*Proof.* Consider the map $f$ and its inverse $f^{-1}$ defined in (10) in the proof of Proposition 4.1. Let $g = f^{-1}(W_1, W_2)$ and $g' = f^{-1}(W_1', W_2')$. By Corollary A.2, since $(W_1, W_2)$ and $(W_1', W_2')$ are not in the same connected component of $L^{-1}(0)$, $g$ and $g'$ are not in the same connected component of $GL_h$. Equivalently, $det(gg') < 0$. Consider a $g_1 \in GL_h$ such that $det(g) < 0$. Then $det(g_1gg') > 0$, which means that $g_1g$ and $g'$ belong to the same connected component of $GL_h$. Therefore, according to Corollary A.2, $g_1 \cdot (W_1, W_2) = f(g_1g)$ and $(W_1', W_2') = f(g')$ belong to the same connected component of $L^{-1}(0)$. $\qquad\square$

**Example.** Suppose $\left(W_1 = \begin{bmatrix} 1 & 0 \\ 0 & 1 \end{bmatrix}, W_2 = \begin{bmatrix} -1 & 0 \\ 0 & 1 \end{bmatrix}\right)$ is a point in $L^{-1}(0)$ for some loss function $L$. Then $\left(W_1' = \begin{bmatrix} -1 & 0 \\ 0 & 1 \end{bmatrix}, W_2' = \begin{bmatrix} 1 & 0 \\ 0 & 1 \end{bmatrix}\right)$ is also a point in $L^{-1}(0)$. However, $(W_1, W_2)$ and $(W_1', W_2')$ are not on the same connected component of the minimum, since their determinants have different signs. By Lemma 5.1, any $g \in GL(h)$ with $det(g) < 0$ can bring $(W_1, W_2)$ and $(W_1', W_2')$ to the same connected component in $L^{-1}(0)$. Let $g$ be the permutation matrix $\begin{bmatrix} 0 & 1 \\ 1 & 0 \end{bmatrix}$. Then $g \cdot (W_1, W_2) = \left(\begin{bmatrix} 0 & 1 \\ 1 & 0 \end{bmatrix}, \begin{bmatrix} 0 & -1 \\ 1 & 0 \end{bmatrix}\right)$, which is in the same connected component as $(W_1', W_2')$.

**Proposition 5.2.** *Assume that $h \geq 2$. For all $(W_1, ..., W_l), (W_1', ..., W_l') \in L^{-1}(0)$, these exists a list of permutation matrices $P_1, ..., P_{l-1}$ such that $(W_1 P_1, P_1^{-1} W_2 P_2, ..., P_{l-2} W_{l-1} P_{l-1}, P_{l-1} W_l)$ and $(W_1', ..., W_l')$ are connected in $L^{-1}(0)$.*

*Proof.* Let $(g_1, ..., g_{l-1}), (g_1', ..., g_{l-1}') \in (GL_h)^{n-1}$ such that $f(g_1, ..., g_{l-1}) = (W_1, ..., W_l)$ and $f(g_1', ..., g_{l-1}') = (W_1', ..., W_l')$. Let $P_0 = I$. For $i = 1, ..., l - 1$, if $det(g_i g_i' P_{i-1}^{-1}) > 0$, set $P_i$ to $I$. Otherwise, we set $P_i$ to an arbitrary element in $P \in S_h \setminus A_h$, which is not empty when $h \geq 2$.

Let $(g_1'', ..., g_{l-1}'') \in (GL_h)^{n-1}$ such that $f(g_1'', ..., g_{l-1}'') = (W_1 P_1, P_1^{-1} W_2 P_2, ..., P_{l-2} W_{l-1} P_{l-1}, P_{l-1} W_l)$. By the way we construct $P_i$'s, we have $g_i'' = P_{i-1}^{-1} g_i' P_i$ and $det(g_i g_i'') > 0$. Therefore, $g_i$ and $g_i''$ belong to the

same connected component of $(GL_h)^{l-1}$ for all $i$. Since $f$ is a homeomorphism between $(GL_h)^{l-1}$ and $L^{-1}(0)$, $(W_1P_1, P^{-1}W_2P_2, ..., P_{l-2}W_{l-1}P_{l-1}, P_{l-1}W_l)$ and $(W'_1, ..., W'_l)$ are connected in $L^{-1}(0)$. $\qquad\square$

**Proposition 5.3.** *Consider the loss function of the following form*

$$L: \; Param \to \mathbb{R}, W = (W_1, ..., W_l) \mapsto ||Y - W_l\sigma(W_{l-1}f(W_{l-2}, W_{l-3}, ..., W_1, X))||_2^2, \tag{13}$$

*where $f$ is a function of $W_{l-2}, W_{l-3}, ..., W_1, X$, and $\sigma(cz) = c^k\sigma(z)$ for all $c \in \mathbb{R}$ and some $k > 0$. Assume that $||Y||_2 \neq 0$ and $L^{-1}(0) \neq \emptyset$. Also assume that $l \geq 2$. For any positive number $b > 0$, there exist $W, W' \in L^{-1}(0)$ that belong to the same connected component of $L^{-1}(0)$ and $0 < \alpha < 1$, such that $L((1-\alpha)W + \alpha W') > b$.*

*Proof.* Let $W = (W_l, ..., W_2, W_1) \in L^{-1}(0)$ be an arbitrary point on the minimum of $L$. Let $W' = (W'_l, ..., W'_2, W'_1) = (W_l m^{-k}, mW_{l-1}, W_{l-2}, ..., W_1)$. Then $W, W'$ belong to the same connected component of $L^{-1}(0)$, connected by curve $\gamma: \mathbb{R} \to Param, \gamma(t) = ((1-t)W_l + tW_l m^{-k}, (1-t)W_{l-1} + tmW_{l-1}, W_{l-2}, ..., W_1)$.

Since $W \in L^{-1}(0)$, we have $W_l\sigma[W_{l-1}f(W_{l-2}, ..., W_1, X)] = Y$. The loss on the linear interpolation of $W, W'$ is

$$\begin{aligned}
L((1-\alpha)W + \alpha W') &= ||Y - ((1-\alpha)W_l + \alpha W'_l)\sigma[((1-\alpha)W_{l-1} + \alpha W'_{l-1})f(W_{l-2}, ..., W_1, X)]||_2^2 \\
&= ||Y - (1-\alpha+\alpha m^{-k})W_l\sigma[(1-\alpha+\alpha m)W_{l-1}f(W_{l-2}, ..., W_1, X)]||_2^2 \\
&= ||Y - (1-\alpha+\alpha m^{-k})(1-\alpha+\alpha m)^k W_l\sigma[W_{l-1}f(W_{l-2}, ..., W_1, X)]||_2^2 \\
&= (1 - (1-\alpha+\alpha m^{-k})(1-\alpha+\alpha m)^k)^2||Y||_2^2. \tag{14}
\end{aligned}$$

Let $\alpha = 0.5$. Then

$$\begin{aligned}
L((1-\alpha)W + \alpha W') &= \left(1 - \left(\frac{1}{2} + \frac{1}{2}m^{-k}\right)\left(\frac{1}{2} + \frac{1}{2}m\right)^k\right)^2 ||Y||_2^2 \\
&= \left(1 - 2^{-(k+1)}(1 + m^{-k})(1 + m)^k\right)^2 ||Y||_2^2 \tag{15}
\end{aligned}$$

Let $m = \left(2^{k+1}\left(\frac{\sqrt{b}}{||Y||^2} + 1\right) - 1\right)^k$. Recall that $k > 0$. Then $m > 0$, $(1+m)^k > 1$, and

$$2^{-(k+1)}(1 + m^{-k})(1 + m)^k > 2^{-(k+1)}(1 + m^{-k}) = \frac{\sqrt{b}}{||Y||^2} + 1 > 1. \tag{16}$$

Therefore, the loss at our chosen values of $\alpha$ and $m$ is at least $b$:

$$L((1-\alpha)W + \alpha W') > \left(1 - \left(\frac{\sqrt{b}}{||Y||^2} + 1\right)\right)^2 ||Y||_2^2 = b. \tag{17}$$

$\qquad\square$

Figure 4 visualizes the loss barrier on the linear interpolation between two minima. We construct a network with loss function $||W_5\sigma(W_4\sigma(W_3\sigma(W_2\sigma(W_1X)))) - Y||$, with $\sigma$ being a leaky ReLU function, $X \in \mathbb{R}^{8\times 4}, Y \in \mathbb{R}^{4\times 4}$, and $(W_1, W_2, W_3, W_4, W_5) \in Param = \mathbb{R}^{16\times 8} \times \mathbb{R}^{32\times 16} \times \mathbb{R}^{16\times 32} \times \mathbb{R}^{8\times 16} \times \mathbb{R}^{4\times 8}$. The network is initialized with random weights, and each element of $X, Y$ is sampled independently from a normal distribution.

We obtain the first minima $(W'_1, W'_2, W'_3, W'_4, W'_5)$ by SGD, and the second $(W''_1, W''_2, W''_3, W''_4, W''_5) = (W'_1, W'_2, W'_3, mW'_4, W'_5m^{-1})$ by rescaling the last two layers with $m \in \mathbb{R}^+$. At large $m$, the two minima are farther apart, and the loss evaluated at the middle point of their linear interpolation grows unboundedly as predicted by Proposition 5.3.

**Proposition 5.4.** *Consider the loss function with the same set of assumptions in Proposition 5.3. Assume additionally that there does not exist a permutation $P$ such that every column of $P\sigma(W_{l-1}f(W_{l-2}, W_{l-3}, ..., W_1, X))$ is in the null space of $W_l$. For any positive number $b > 0$, there exist $(W_1, ..., W_l), (W'_1, ..., W'_l) \in L^{-1}(0)$ and $0 < \alpha < 1$, such that $(W_1, ..., W_{l-2}) = (W'_1, ..., W'_{l-2})$ and $\min_{P \in S_n} L((1-\alpha)(W_1, ..., W_l) + \alpha(W_1, ..., W_{l-2}, P^{-1}W_{l-1}, W_lP)) > b$.*

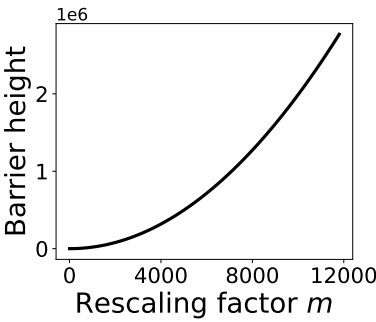

Figure 4: Loss at the middle of the linear interpolation between two minima in a homogeneous network becomes unbounded when the two minima is far apart.

*Proof.* Let $W = (W_l, ..., W_2, W_1) \in L^{-1}(0)$ be an arbitrary point on the minimum of $L$. Let $W' = (W'_l, ..., W'_2, W'_1) = (W_l m^{-k}, m W_{l-1}, W_{l-2}, ..., W_1)$.

Since $W \in L^{-1}(0)$, we have $W_l \sigma [W_{l-1} f(W_{l-2}, ..., W_1, X)] = Y$. The loss on the linear interpolation of $W, W'$ is

$$L((1-\alpha)W + \alpha W') = ||Y - ((1-\alpha)W_l + \alpha W'_l P)\sigma[((1-\alpha)W_{l-1} + \alpha P^{-1}W'_{l-1})f(W_{l-2}, ..., W_1, X)]||_2^2. \quad (18)$$

Let $\alpha = 0.5$. Then

$$L((1-\alpha)W + \alpha W') = ||Y - \frac{1}{4}W_l(I + m^{-k}P)\sigma[(I + mP^{-1})W_{l-1}f(W_{l-2}, ..., W_1, X)]||_2^2. \quad (19)$$

When $m \to \infty$,

$$\lim_{m \to \infty} \sigma[(I + mP^{-1})W_{l-1}f(W_{l-2}, ..., W_1, X)]$$
$$= \lim_{m \to \infty} m^k \sigma[(m^{-1}I + P^{-1})W_{l-1}f(W_{l-2}, ..., W_1, X)]$$
$$= \lim_{m \to \infty} m^k P^{-1}\sigma[W_{l-1}f(W_{l-2}, ..., W_1, X)]. \quad (20)$$

Therefore,

$$\lim_{m \to \infty} L((1-\alpha)W + \alpha W') = \lim_{m \to \infty} ||Y - \frac{1}{4}W_l(I + m^{-k}P)m^k P^{-1}\sigma[W_{l-1}f(W_{l-2}, ..., W_1, X)]||_2^2$$
$$= \lim_{m \to \infty} ||Y - \frac{1}{4}W_l(I + m^k P^{-1})\sigma[W_{l-1}f(W_{l-2}, ..., W_1, X)]||_2^2$$
$$= \lim_{m \to \infty} ||\frac{3}{4}Y - \frac{m^k}{4}W_l P^{-1}\sigma[W_{l-1}f(W_{l-2}, ..., W_1, X)]||_2^2. \quad (21)$$

Since we assumed that there does not exist a permutation $P$ such that every column of $P\sigma(W_{l-1}f(W_{l-2}, W_{l-3}, ..., W_1, X))$ is in the null space of $W_l$, at least one element in the second term is unbounded for any permutation $P$. Therefore, $L((1-\alpha)W + \alpha W')$ is unbounded for any $P$. □

**Proposition 5.5.** *Let $A \in \mathbb{R}^{n \times n}$ be an invertible matrix. Let set $S = \{(W_1, W_2) : W_1, W_2 \in \mathbb{R}^{n \times n}, W_1 W_2 = A\}$. For any positive number $b > 0$, there exist $W', W'' \in S$ and $0 < \alpha < 1$, such that $\min_{\hat{W} \in S} ||((1-\alpha)W' + \alpha W'') - \hat{W}||_2 > b$.*

*Proof.* Let $W$ be an element of $S$. Let $W'_1 = W_1 g_1^{-1}$, $W'_2 = g_1 W_2$, $W''_1 = W_1 g_2^{-1}$, and $W''_2 = g_2 W_2$, where $g_1, g_2 \in \mathbb{R}^{n \times n}$ are invertible matrices. Note that $W' = (W'_1, W'_2)$ and $W'' = (W''_1, W''_2)$ are both in $S$. Then,

$$\min_{\hat{W} \in S} ||((1-\alpha)W' + \alpha W'') - \hat{W}||_2^2$$
$$= \min_{\hat{W} \in S} ||(1-\alpha)W_1 g_1^{-1} + \alpha W_1 g_2^{-1} - \hat{W}_1||_2^2 + ||(1-\alpha)g_1 W_2 + \alpha g_2 W_2 - \hat{W}_2||_2^2$$
$$= \min_{g \in GL(n)} ||W_1((1-\alpha)g_1^{-1} + \alpha g_2^{-1} - g^{-1})||_2^2 + ||W_2((1-\alpha)g_1 + \alpha g_2 - g)||_2^2. \quad (22)$$

Let $g_1 = \beta I$ and $g_2 = \beta^{-1} I$ for some $\beta > 0$. Let $\alpha = \frac{1}{2}$. Then, in the limit of a large $\beta$, we have

$$\lim_{\beta \to \infty} \min_{\hat{W} \in S} \| ((1-\alpha)W + \alpha W') - \hat{W} \|_2^2$$

$$= \lim_{\beta \to \infty} \min_{g \in GL(n)} \left\| W_1 \left( \frac{\beta + \beta^{-1}}{2} I - g^{-1} \right) \right\|_2^2 + \left\| W_2 \left( \frac{\beta + \beta^{-1}}{2} I - g \right) \right\|_2^2. \tag{23}$$

As $\beta \to \infty$, $g$ and $g^{-1}$ cannot approach $\frac{\beta + \beta^{-1}}{2} I$ simultaneously. Therefore, (23) is not bounded. $\qquad\square$

**Proposition 5.6.** *Consider the loss function with the same set of assumptions in Proposition 5.3. Let $W \in L^{-1}(0)$ be a point on the minimum. Consider the multiplicative group of positive real numbers $\mathbb{R}^+$ that acts on $L^{-1}(0)$ by $g \cdot (W_1, ..., W_l) = (W_1, ..., W_{l-2}, gW_{l-1}, W_l g^{-k})$, where $g \in \mathbb{R}^+$. Then there exists a positive number $b > 0$, such that for all $0 < \alpha < 1$ and $W' \in Orbit(W)$ with $\|W_i'\|_2 < c$ for all $i$ and some $c > 0$, the loss value for points on the linear interpolation $L((1-\alpha)W + \alpha W') < b$.*

*Proof.* Since $W' \in Orbit(W)$, $W' = (W_l m^{-k}, mW_{l-1}, W_{l-2}, ..., W_1)$ for some $m > 0$. Additionally, $m$ and $m^{-k}$ are bounded since $W_i'$ is bounded. Since $W \in L^{-1}(0)$, we have $W_l \sigma [W_{l-1} f(W_{l-2}, ..., W_1, X)] = Y$. The loss on the linear interpolation of $W, W'$ is

$$\begin{aligned}
L((1-\alpha)W + \alpha W') &= \|Y - ((1-\alpha)W_l + \alpha W_l')\sigma [((1-\alpha)W_{l-1} + \alpha W_{l-1}')f(W_{l-2}, ..., W_1, X)] \|_2^2 \\
&= \|Y - (1 - \alpha + \alpha m^{-k})W_l \sigma [(1 - \alpha + \alpha m)W_{l-1}f(W_{l-2}, ..., W_1, X)] \|_2^2 \\
&= \|Y - (1 - \alpha + \alpha m^{-k})(1 - \alpha + \alpha m)^k W_l \sigma [W_{l-1}f(W_{l-2}, ..., W_1, X)] \|_2^2 \\
&= (1 - (1 - \alpha + \alpha m^{-k})(1 - \alpha + \alpha m)^k)^2 \|Y\|_2^2.
\end{aligned} \tag{24}$$

As $m$, $m^{-k}$, and $\alpha$ are all bounded, the loss value for points on the linear interpolation $L((1-\alpha)W + \alpha W')$ is also bounded. $\qquad\square$

The connectedness results derived from symmetry raise several interesting questions about mode connectivity. For example, it would be interesting to understand when and why there is no significant change in loss on the linear interpolation between two minima. One possible explanation is that there always exists a symmetry-induced path $\gamma$ that stays close to the linear interpolation. Another potential factor is the high dimensionality of the minimum, which increases the likelihood that a significant portion of the linear interpolation remains within the low-loss region. Additionally, empirical observations suggest that both train and test accuracy remain nearly constant along paths connecting different SGD solutions (Garipov et al., 2018). If these paths are induced by a group action, this would imply that the group action's dependence on data is weak. Investigating the extent to which data influences these symmetries could provide deeper insights into the structure of the loss landscape and the generalization properties of neural networks.

## E. Proofs in Section 6

**Proposition 6.1.** *Let $(U, V) \in Param$, and $(U', V') = g \cdot (U, V)$. Then*

$$\|U\sigma(VX) - U'\sigma(V'X)\| \leq \|U\sigma(VX)\|. \tag{25}$$

*Proof.* We note that $I - \sigma(gVX)^\dagger \sigma(gVX)$ is a projection:

$$\begin{aligned}
&(I - \sigma(gVX)^\dagger \sigma(gVX))^2 \\
&= I - \sigma(gVX)^\dagger \sigma(gVX) - \sigma(gVX)^\dagger \sigma(gVX)(I - \sigma(gVX)^\dagger \sigma(gVX)) \\
&= I - \sigma(gVX)^\dagger \sigma(gVX).
\end{aligned}$$

Therefore,

$$\|U\sigma(VX) - U'\sigma(V'X)\| = \|U\sigma(VX)\left(I - \sigma(gVX)^\dagger \sigma(gVX)\right)\| \leq \|U\sigma(VX)\|. \tag{26}$$

$\qquad\square$

**Theorem 6.2.** *Let $L^{-1}(c) \subset$ Param, with $c \in \mathbb{R}$, be a level set of the loss function $L :$ Param $\to \mathbb{R}$. Let $\gamma : [0,1] \to L^{-1}(c)$ be a smooth curve in $L^{-1}(c)$ connecting two points $\boldsymbol{w}_1 = \gamma(0)$ and $\boldsymbol{w}_2 = \gamma(1)$. Suppose the curvature $\kappa(t)$ of $\gamma$ satisfies $\kappa(t) \leq \kappa_{\max}$ for all $t \in [0,1]$.*

*Let $S$ be the straight line segment connecting $\boldsymbol{w}_1$ and $\boldsymbol{w}_2$. Then, for any point $\boldsymbol{w}$ on $S$, the distance to $L^{-1}(c)$ is bounded by*

$$\text{dist}(\boldsymbol{w}, L^{-1}(c)) \leq d_{\max} = \frac{1}{\kappa_{\max}} \left( 1 - \sqrt{1 - \left( \frac{\kappa_{\max} \| \boldsymbol{w}_2 - \boldsymbol{w}_1 \|_2}{2} \right)^2} \right).$$

*Furthermore, assuming $L$ is Lipschitz continuous with Lipschitz constant $C_L$, the loss at any point $\boldsymbol{w}$ on $S$ satisfies*

$$|L(\boldsymbol{w}) - c| \leq C_L d_{\max}.$$

*Proof.* We will find an upper bound for the maximum distance between a smooth curve and the chord connecting two points on the curve, assuming the curvature of the curve is bounded by $\kappa_{\max}$.

The curvature $\kappa$ at a point on a curve is defined as $\kappa = \frac{1}{R}$, where $R$ is the radius of the osculating circle at that point. Let $s$ be the maximum perpendicular distance from the midpoint of a chord to the curve. For a circular arc, Pythagorean theorem gives

$$R^2 = \left( \frac{\| \boldsymbol{w}_2 - \boldsymbol{w}_1 \|_2}{2} \right)^2 + (R - s)^2.$$

Solving for $s$:

$$s = R \left( 1 - \sqrt{1 - \left( \frac{\| \boldsymbol{w}_2 - \boldsymbol{w}_1 \|_2}{2R} \right)^2} \right).$$

Substitute $R = \frac{1}{\kappa}$ into the above, we have

$$s = \frac{1}{\kappa} \left( 1 - \sqrt{1 - \left( \frac{\kappa \| \boldsymbol{w}_2 - \boldsymbol{w}_1 \|_2}{2} \right)^2} \right).$$

Since the curvature of $\gamma$ is everywhere less than or equal to $\kappa_{\max}$, the curve cannot bend more sharply than the osculating circle with curvature $\kappa_{\max}$. Therefore, the maximum deviation $d_{\max}$ between $\gamma$ and its chord cannot exceed that of the osculating circle:

$$\text{dist}(\boldsymbol{w}, L^{-1}(c)) \leq d_{\max} \overset{\text{def}}{=} \frac{1}{\kappa_{\max}} \left( 1 - \sqrt{1 - \left( \frac{\kappa_{\max} \| \boldsymbol{w}_2 - \boldsymbol{w}_1 \|_2}{2} \right)^2} \right).$$

Assuming $L$ is Lipschitz continuous with Lipschitz constant $C_L$, for any $\boldsymbol{w}$ on $S$, we have

$$|L(\boldsymbol{w}) - c| = |L(\boldsymbol{w}) - L(\gamma(t))| \leq C_L \| \boldsymbol{w} - \gamma(t) \| \leq C_L d_{\max}.$$

$\square$

