# OpenReview forum: "Understanding Mode Connectivity via Parameter Space Symmetry"
_ICML.cc/2025/Conference — ICML 2025 poster_

### Official Review · Reviewer_ZqPy · 2025-03-13

**Overall Recommendation:** 4

**Summary:**

The paper investigates (linear) mode connectivity of neural networks via symmetries in parameter space. Besides quantifying the number of connected components of invertible linear networks with and without skip connections, it sheds light on when modes can be connected linearly by investigating the difference of the linear path between two modes and the orbit between the two points.

**Claims And Evidence:**

The paper investigates the connection between symmetries in parameter space and mode connectivity theoretically. All theoretical claims are rigorously proven. Some claims (e.g., connected components w/o skip connections and loss preservation) are supported empirically.

**Essential References Not Discussed:**

To the best of my knowledge, the paper references all essential works.

**Experimental Designs Or Analyses:**

As stated above, the main contributions of the paper are theoretical. When empirical evidence was presented, the experimental design was sound.

**Methods And Evaluation Criteria:**

The paper's contributions are theoretical, yet a few claims are supported by empirical evidence. Where this is the case, the methods and criteria are sound.

**Other Comments Or Suggestions:**

- It is a bit of a long shot, but in learning theory for neural networks, one problem is that some generalization measures change through reparameterizations. Therefore, reparameterization-invariant measures have been proposed [4,5] and some can even be theoretically connected to generalization. Would, for example the proposed relative flatness [5] be invariant under the reparameterizations discussed in this paper?
- Prop. 5.4: in the equivalence of weights $W_i$ and $W'_i$ at the end of the paragraph, the permutation $P$ seems to be missing.

[4] Tsuzuku, Yusuke, Issei Sato, and Masashi Sugiyama. "Normalized flat minima: Exploring scale invariant definition of flat minima for neural networks using pac-bayesian analysis." International Conference on Machine Learning. PMLR, 2020.
[5] Petzka, Henning, et al. "Relative flatness and generalization." Advances in neural information processing systems 34 (2021): 18420-18432.

######### After Rebuttal ############

I maintain my positive assessment and recommend acceptance.

**Other Strengths And Weaknesses:**

The paper is very well-written, the claims are clearly presented, and the proofs are sound. One thing that could be improved is a discussion on the limitations of the analysis: most of the analysis is for linear neural networks and thus the applicability to modern deep learning is limited. While a detailed empirical analysis of this difference might be out of scope, a discussion of these limitations seems appropriate.

**Questions For Authors:**

Q1: Cor. 4.2 shows that the number of connected components increases with depth, whereas width plays no role. This is interesting. Can you comment on potential implications for neural network training? It seems related to the fact that deeper networks can be trained easier than wide and shallow ones.

**Relation To Broader Scientific Literature:**

- It would be interesting to discuss how the studied symmetries of the general linear group relate to the symmetries discussed by Petzka et al. [1].
- How do the loss-preserving orbits discussed in this paper relate to the minima manifold in [2]?
- This paper discusses linear reparameterizations. How does it relate to works on non-linear reparameterizations using Riemannian geometry [3]?

[1] Petzka, Henning, Martin Trimmel, and Cristian Sminchisescu. "Notes on the symmetries of 2-layer relu-networks." Proceedings of the northern lights deep learning workshop. Vol. 1. 2020.
[2] Simsek, Berfin, et al. "Geometry of the loss landscape in overparameterized neural networks: Symmetries and invariances." International Conference on Machine Learning. PMLR, 2021.
[3] Kristiadi, Agustinus, Felix Dangel, and Philipp Hennig. "The geometry of neural nets' parameter spaces under reparametrization." Advances in Neural Information Processing Systems 36 (2023): 17669-17688.

**Theoretical Claims:**

I have checked the theoretical claims up to my abilities. The proofs presented in the appendix are clear and sound.

---

> ### Author Rebuttal · Authors · 2025-04-01
>
> We thank the reviewer for their encouraging feedback. We appreciate that they have taken the time to read our proofs. We also appreciate the valuable questions and the many relevant pointers to related work, which we discuss below.
>
> **Relation to broader scientific literature**
>
> > It would be interesting to discuss how the studied symmetries of the general linear group relate to the symmetries discussed by Petzka et al. [1].
>
> Petzka et al. show that permutation and rescaling are the only transformations that leave a 2-layer ReLU network unchanged, excluding degenerate cases. These symmetries are subgroups of the general linear symmetry group studied in our work. They also identify degenerate-case transformations not present in linear networks, when neurons share identical zero hyperplanes.
>
> > How do the loss-preserving orbits discussed in this paper relate to the minima manifold in [2]?
>
> Both our paper and Simsek et al. [2] analyze symmetry-induced structures in the loss landscape. Our orbits arise from continuous symmetry groups (e.g., GL(n), rescalings), while their minima manifold arises by embedding smaller networks into overparameterized ones via permutations and neuron duplications. These manifolds can be viewed as unions of affine subspaces, and our orbits form structured subsets within it. The two views are complementary and compatible, especially in linear networks with full-rank weights.
>
> > This paper discusses linear reparameterizations. How does it relate to works on non-linear reparameterizations using Riemannian geometry [3]?
>
> Kristiadi et al. [3] study reparameterization as a map from the parameter space to a new space, while we focus on symmetry that maps from the parameter space to itself. Both maps can be nonlinear. Studying symmetry-induced orbits from a Riemannian perspective could be a promising direction for future work.
>
> We will include a brief discussion of these connections in the final version of the paper.
>
>
> **Limitation of the analysis**
>
> > most of the analysis is for linear neural networks and thus the applicability to modern deep learning is limited. While a detailed empirical analysis of this difference might be out of scope, a discussion of these limitations seems appropriate.
>
> We appreciate the suggestion and agree that a discussion would be valuable. Our results are derived under architectural assumptions that enable precise, mathematically grounded insights into how continuous symmetries shape the loss landscape.
>
> While the findings may not directly transfer to all architectures, our methods are adaptable. For example, Section 5.3 applies to any network with layers involving multiplication of two weight matrices–a pattern that appears, for example, in transformer blocks. In higher-dimensional, non full-rank settings, connectivity depends on how orbits defined by different rank combinations intersect. Extensions to nonlinear networks are possible by identifying approximate symmetries (as in Section 6), or by continuously deforming the minima and studying its behavior in the limit as the network approaches a linear regime.Even partial or approximate symmetry can provide useful structural information about the minima and support new applications.
>
> **Other comments and questions**
>
> > Would, for example the proposed relative flatness [5] be invariant under the reparameterizations discussed in this paper?
>
> Yes, in many cases. The relative flatness measure proposed in Petzka et al. [5] is designed to be invariant under layer-wise and neuron-wise linear reparameterizations, such as rescaling and certain changes of basis. These are precisely the types of continuous architectural symmetries we analyze in our paper. Therefore, for the linear and homogeneous network settings considered in our work, the relative flatness measure would indeed remain invariant under the reparameterizations we describe.
>
> > Prop. 5.4: in the equivalence of weights $W_i$ and $W_i'$ at the end of the paragraph, the permutation $P$ seems to be missing.
>
> This is indeed a typo. We will add the permutations. Thank you for the detailed read.
>
> > Q1: Cor. 4.2 shows that the number of connected components increases with depth, whereas width plays no role. This is interesting. Can you comment on potential implications for neural network training? It seems related to the fact that deeper networks can be trained easier than wide and shallow ones.
>
> We do not have concrete results on whether more connected components correlates with easier training, but below we share some speculations. A larger number of connected components could mean the minima is distributed more widely in the parameter space, which makes them easier to find during training. However, an exact analysis would require quantifying the area (or volume) and the exact distribution of the minima, as well as the geometry in the surrounding region. Nevertheless, connecting connectedness and optimization seems to be an exciting line of future work.

---

> > ### Comment · Reviewer_ZqPy · 2025-04-02
> >
> > I would like to thank the authors for their rebuttal. They have answered my questions and clarified several points. I think this is a good paper and therefore maintain my positive rating.

---

### Official Review · Reviewer_GSK8 · 2025-03-14

**Overall Recommendation:** 3

**Summary:**

This paper provides a group theory framework to study the connectivity and the number of components of the zero-loss set in a (non-convex) loss landscape. A precise characterization of the number of components and connectivity of the zero-loss set is given for deep linear neural networks. Some results on the existence of a high-loss barrier are given for non-linear neural networks.

**Claims And Evidence:**

It is fairly clear. Some well-known simple things are presented as Propositions (for ex. Propositions 3.4 and 3.5). These are not particular contributions of this paper hence should not be presented as Propositions.

**Essential References Not Discussed:**

No.

**Experimental Designs Or Analyses:**

NA.

**Methods And Evaluation Criteria:**

NA

**Other Comments Or Suggestions:**

```Param``` should be called $\Theta$.

**Other Strengths And Weaknesses:**

Strength: Characterization of the zero-loss set of deep linear networks is novel and group-theoretic background is the way forward.

Weakness: ResNet 1D section should not be called that as that architecture has no non-linearity.

**Questions For Authors:**

Comment: Proposition 5.2 uses permutations to connect components for a deep ---linear--- network. However, the relevant symmetry group here is $GL_n$ as those matrices can be multiplied by any invertible matrix without changing the loss. In other words, symmetries of deep linear networks have more ---volume--- compared to the deep non-linear networks.

Please add a comment along these lines so that the readers are aware of this distinction between linear and non-linear networks.

**Relation To Broader Scientific Literature:**

Linear mode connectivity of deep neural networks is an interesting and active area of research with implications for mode connectivity and model merging. A complete and group-theory-based characterization of the zero-loss set of parameters for deep linear networks is a nice contribution to the literature (but I have some reservations about the symmetry-group relevant for deep linear networks).

Moreover, the existence of a big barrier is characterized. This is expected for the distant components which would be rare compared to the close components which would be more common. SGD likely finds more common components which explain the observed phenomenon of linear connectivity.

**Theoretical Claims:**

I did not find a mistake in Theorems.

---

> ### Author Rebuttal · Authors · 2025-04-01
>
> We thank the reviewer for their encouraging feedback and insights on our work’s relation to broader literature. We will incorporate their suggestions into the final version of the paper.
>
> > Some well-known simple things are presented as Propositions (for ex. Propositions 3.4 and 3.5). These are not particular contributions of this paper hence should not be presented as Propositions.
>
> Section 3 is intended as a background section, and it is not our intention to present these propositions as novel results. We chose to include them in proposition form primarily for completeness and ease of referencing them in proofs. We agree they should not be positioned as contributions of this paper, and we will revise the paper to clarify this.
>
> > ResNet 1D section should not be called that as that architecture has no non-linearity.
>
> Thank you for pointing this out. We will change the terminology to skip-connection to avoid ambiguity.
>
> > Comment: Proposition 5.2 uses permutations to connect components for a deep ---linear--- network. However, the relevant symmetry group here is $GL_n$ as those matrices can be multiplied by any invertible matrix without changing the loss. In other words, symmetries of deep linear networks have more ---volume--- compared to the deep non-linear networks.
>
> > Please add a comment along these lines so that the readers are aware of this distinction between linear and non-linear networks.
>
> We appreciate this observation and will include a comment along these lines. While permutations are sufficient to connect components in deep linear networks, the full symmetry group in this case is indeed $GL_n$. Also, it is indeed true that deep linear networks have larger symmetry groups than deep non-linear networks.

---

### Official Review · Reviewer_MTSB · 2025-03-15

**Overall Recommendation:** 3

**Summary:**

This work provides an interesting perspective on mode connectivity by linking topology of symmetry groups to the topology of minima. The key technique used in the paper is based on deriving the number of connected components of minima in linear networks (showing $2^{l-1}$ components for a network with $l$ layers), and in addition they show that skip connections reduce this number (although this is done in a very simplified setting,  e.g., from 4 to 3 in a scalar ResNet). Based on this the authors go on to prove mode connectivity up to permutation for linear two-layer networks with invertible weights, and then go on to show how this fails when considering the multi-layer case. Further, some interesting analysis is carried to find curves on the minimum

**Claims And Evidence:**

Yes, more or less. But see the later discussion in questions or weaknesses, where there are some questions on how these results fit in to the empirical behaviour of LMC.

**Essential References Not Discussed:**

n/a

**Experimental Designs Or Analyses:**

See the strengths and weaknesses

**Methods And Evaluation Criteria:**

See the strengths and weaknesses

**Other Comments Or Suggestions:**

n/a

**Other Strengths And Weaknesses:**

Strengths:
- The theoretical connections made are quite interesting and might lead to more such works from the topological perspective.
- The accompanying setups and figures provide useful intuition to think about the problem.
- The paper is quite well written.

Weaknesses:
- Despite the above, the practical takeaways practitioners or those working in the area can have remain vague at best.

**Questions For Authors:**

- Is it clear that having a lower number of connected components makes for easier connectivity? I am not so sure, as even in a single connected space, it might require a complicated path. Maybe it would be nice to have some sort of discussion in the paper, as what would be ideal atleast from a topological perspective to connect the network solutions.

- How do you reconcile the failure case of multi-layers with the empirically observed largely well-connected paths for deep networks?

- I am curious what are the nature of theoretical bottlenecks that impede a general analysis for ResNets, besides the general case.

**Relation To Broader Scientific Literature:**

The paper builds on and extends prior work on mode connectivity and loss landscapes. The key contribution—linking symmetry group topology to minima structure—is novel and brings in a complementary and rigorous perspective.

**Theoretical Claims:**

no

---

> ### Author Rebuttal · Authors · 2025-04-01
>
> We thank the reviewer for their constructive feedback and insightful questions. We are encouraged by the recognition of our paper’s novelty, rigor, and intuition. Below we expand on the practical takeaways and respond to the questions.
>
> **Practical takeaways**
>
> Our work shows that parameter space symmetries, especially continuous ones, explain why and how minima are connected. For practitioners, these results motivate concrete strategies – and cautions – for tasks that navigate the loss landscape, including model merging, ensembling, and fine-tuning:
>
> - One can build low-loss curves explicitly using known parameter symmetries. This gives a principled and efficient way to obtain new minima from old ones, potentially useful in (1) generating model ensembles with low cost; (2) improving model alignment by allowing a much larger group of transformations than permutation; and (3) mitigating catastrophic forgetting in fine-tuning by constraining updates to remain on the symmetry-induced manifold of the pretraining minimum.
>
> -  The connectedness of minima supports the practice of model merging and ensembling, even when models are trained separately. In addition to permutation, many other symmetry transformations can connect solutions that would otherwise appear very different.
>
> - Linear interpolation between minima is not guaranteed to lead to better models, despite its widespread use. This highlights the need to evaluate whether the minima found by specific learning algorithms are approximately connected before averaging models directly.
>
> Finally, for theorists working in this area, shifting focus from viewing loss landscapes as chaotic to structured, symmetry-rich spaces invites new mathematical tools to explain empirical observations in deep learning. This has led to new intuitions ranging from why residual connections lead to more connected minima, why linear model connectivity can fail, to why model averaging often works when the models are not too far from each other. We also hope that the broader approach – inferring properties of unknown solution sets from known symmetry groups – could inspire new work beyond this field.
>
> **Response to questions**
>
> > Is it clear that having a lower number of connected components makes for easier connectivity?
>
> We agree with the reviewer that connectedness alone does not imply easy connectivity in the sense of short or simple paths between solutions. Being in the same connected components is a necessary condition for connectivity, but a single component may still contain complex geometry necessitating complicated connecting paths.
>
> Defining the ease of connectivity is subtle, and we agree that a discussion would be valuable. One natural measure is the parametric complexity of the connecting curves, quantifiable by their degree if polynomial, or number of segments if piece-wise. Another possible definition for easy connectivity would be low curvature of the minimum manifold or short geodesic distance between two points on it. As we discussed in Section 6.2, low curvature implies that linear interpolation stays near the manifold. Other potential definitions include whether the connecting curve has an analytical expression, or how many points are needed to approximate it within a certain error. It would indeed be interesting to examine these properties for symmetry-induced connecting curves.
>
> > How do you reconcile the failure case of multi-layers with the empirically observed largely well-connected paths for deep networks?
>
> The empirical observation of mode connectivity and linear mode connectivity is likely due to the fact that certain family of optimizers, especially stochastic gradient descent (SGD), tend to explore only a subset of the minimum, a phenomenon often referred to as implicit bias. Regularization techniques and weight decay may further encourage SGD to favor certain regions of the minimum. As a result, the subset of minima that is likely to be reached by SGD can have very different structures and perhaps more connected than the full set of minima.
>
> > I am curious what are the nature of theoretical bottlenecks that impede a general analysis for ResNets, besides the general case.
>
> Extending our proof to a general analysis is challenging because the number of orbits grows rapidly with the dimension of the weight matrices. In the 1D case, the minimum consists of only two orbits, so the proof only requires analyzing the connected components of each and checking how they intersect. For higher-dimensional weights, especially when not full-rank, each distinct rank combination defines multiple orbits, and characterizing the intersections among these many orbits, while possible, becomes combinatorially complex. We therefore chose to include only the 1D ResNet example to demonstrate the proof technique. It is possible that future work with more refined tools could extend this analysis to higher-dimensional settings more efficiently.

---

### Official Review · Reviewer_7vQu · 2025-03-15

**Overall Recommendation:** 2

**Summary:**

The paper studies the topology of the minima in neural networks through their symmetries. Results of the paper consists of different simplified models of neural networks, for example using single units and single data point, or assuming linear networks. The paper the studies the effect of different types of symmetries – e.g., GL(n) for linear networks, or rescaling symmetry for homogeneous activations – on various properties of the loss landscape – e.g., the number of connected components, linear mode connectivity – in these simplified settings.

**Claims And Evidence:**

Theoretical claims are supported by proofs in the appendix and enough explanation in the main body to give an idea as to why they are correct. Sometimes the scope of the findings is inaccurate when a high-level picture is given; for example, the paper says that it shows the number of connected components for full-rank linear regression is reduced through skip-connection, but this result is limited to 3-layer networks with single neuron in every layer.

**Essential References Not Discussed:**

Not to my knowledge.

**Experimental Designs Or Analyses:**

Experiments are justifiably quite limited.

**Methods And Evaluation Criteria:**

The investigation relies on the symmetries of simplified models of neural networks; for example, the fact that when we remove all activation functions, neural networks have GL(n) symmetry, and the fact this group has two connected components and it subsumes the permutation symmetry. This limits the scope of finding to toy settings where we know for a fact the conclusions break down when we move to a simple MLP. Nevertheless, the paper has some interesting insights, for example, the fact that the topology of the symmetry group affects the topology of the loss level-sets.

**Other Comments Or Suggestions:**

I got lost at the beginning of section 6, after reading equation 6 and proposition 6.1. It would be nice to clarify the material of that section, in particular, why such an arbitrary transformation is used, and what it has to do with a one-parameter subgroup.

**Other Strengths And Weaknesses:**

Strengths

– Nice coverage of related literature

– Clear presentation of necessary background on topology and groups

– Clear statement of results in each section (with the exception of section 6)

Weaknesses

– Results rely on strong assumptions, so much so that unfortunately conclusions tell us nothing about realistic neural networks. Note that as opposed to many toy models, here, when dealing with symmetries, one can be sure that conclusions do not transfer to more complicated architectures, since these symmetries do not exist.

– The paper reads like a collection of loosely related results; it includes different toy models used to study different phenomena.

– The conclusions made about the role of symmetry in this paper completely ignores (and eliminates) the effect of overparameterization symmetry, which has a much larger degree of freedom than symmetries of the architecture and potentially a bigger role to play in the geometry of the landscape.

**Questions For Authors:**

Could you please explain if you expect any of the findings to “approximately” translate to a realistic/practical setting of an MLP, or even a linear neural network in the overparameterized regime?

**Relation To Broader Scientific Literature:**

The section on related works has a discussion of prior works on mode connectivity and symmetries of the loss landscape. I found the coverage quite adequate.

**Theoretical Claims:**

No, I did not check the proof.

---

> ### Author Rebuttal · Authors · 2025-04-01
>
> We thank the reviewer for their comments. Below, we address concerns about the scope and generalizability of our results, and clarify how our approach can be extended.
>
> **Scope and generalizability of results**
>
> > Sometimes the scope of the findings is inaccurate when a high-level picture is given…
>
> We will clarify the intended scope to reduce ambiguity. For higher-dimensional invertible weights in the skip-connection example, the number of components reduces further to 2. We are extending the analysis by relaxing the invertibility condition and will include full proofs in the final version.
>
> > The investigation relies on the symmetries of simplified models of neural networks… This limits the scope of finding to toy settings where we know for a fact the conclusions break down when we move to a simple MLP.
>
> While our analysis focuses on simplified models, it does not fundamentally break down for more complex architectures. Many modern architectures retain large spaces of continuous symmetry—for example, our results in Section 5.3 apply broadly to layers involving matrix multiplication, such as transformer blocks. When a homeomorphism exists between the symmetry group and the set of minima, topological properties such as connectedness can be inferred directly. In non-invertible settings, such as with skip connections, connectivity can still be analyzed through how orbits intersect, albeit with more care.
>
> > Results rely on strong assumptions, so much so that unfortunately conclusions tell us nothing about realistic neural networks. …one can be sure that conclusions do not transfer to more complicated architectures, since these symmetries do not exist.
>
> We respectfully disagree that our findings are irrelevant to realistic neural networks. We do not claim direct applicability to all architectures; rather, we aim to isolate the role of symmetry in shaping the loss landscape. Our simplified models allow for precise, mathematically grounded insights into how symmetries–especially continuous ones–can yield connected minima and help explain phenomena like linear mode connectivity.
>
> As explained above, our methods provide a framework that can be adapted to a wide range of architectures. Even when we do not know the full set of symmetry, a subset of symmetry or even approximate symmetry can still give useful information on the structures of minima and inform new applications, as demonstrated in section 6.
>
> > The conclusions made about the role of symmetry in this paper completely ignores (and eliminates) the effect of overparameterization symmetry…
>
> Overparameterization increases the dimension of the minima set, reflecting redundancy or flatness, but does not induce symmetry groups (e.g., permutations or rescalings) in the group-theoretic sense like the architectural symmetries. Thus, our results neither overlook nor contradict existing theory on overparameterization. Exploring interactions between high-dimensional minima and architectural symmetries remains an interesting direction for future work, and our framework may help analyze such cases.
>
> > Could you please explain if you expect any of the findings to “approximately” translate to a realistic/practical setting of an MLP, or even a linear neural network in the overparameterized regime?
>
> Our main multilayer setting (Equation (1)) already allows for overparameterization. While not all MLPs will exhibit the same topology as our simplified models, we believe our approach is generalizable. For higher-dimensional weights, connectivity analysis involves characterizing intersections among multiple orbits defined by distinct rank combinations. Extending our analysis to include nonlinearity could involve approximate symmetries, as in Section 6, or by continuously deforming the minima and studying its behavior in the limit as the network approaches a linearity.
>
> **Writing and presentation**
>
> > The paper reads like a collection of loosely related results; it includes different toy models used to study different phenomena.
>
> The paper's central theme is that parameter space symmetries, especially continuous ones, can help explain why and how minima are connected. Each model is chosen to isolate a specific aspect of this symmetry-connectivity relationship. Collectively, they build a coherent picture of how architectural symmetries shape the loss landscape and empirical behaviors observed in deep learning.
>
> > I got lost at the beginning of section 6... It would be nice to clarify the material of that section…
>
> We will revise this section to clarify and better motivate the use of one-parameter subgroups. The transformation in Equation 6 is not arbitrary–it is specifically constructed to follow a one-parameter subgroup and defines a smooth curve within the minima. The section analyzes the curvature of such symmetry-induced curves to understand when linear mode connectivity approximately holds. Proposition 6.1 formalizes this by relating interpolation loss to curvature.

---

> > ### Comment · Reviewer_7vQu · 2025-04-02
> >
> > Thank you for your response.
> >
> > Could you please clarify the implications of the full-rank assumption about X and Y in your results?

---

> > > ### Author Response · Authors · 2025-04-03
> > >
> > > Thank you for your question. The full-rank assumption ensures that the minimum is non-empty and well-structured. While it simplifies the analysis in some settings, many of our results do not rely on it, and our approach can generally extend beyond this assumption. Specifically:
> > >
> > > - **Sections 5.2, 5.3, and 6 (linear mode connectivity, curves on the minimum):** These results do not require $X$ and $Y$ to be full-rank.
> > >
> > > - **Section 4 (connected components):** Most results here assume full-rank $X$ and $Y$. Without this assumption, each loss level set may consist of more than one orbit under the group action, corresponding to different rank combinations of the weight matrices. The number of connected components may increase or decrease depending on the architecture (see Proposition A.8). In such cases, one can apply the same connectivity analysis, with the added step of characterizing intersections among multiple orbits.
> > >
> > > - **Section 5.1 (permutation-induced connectivity):** In their current form, these results assume full-rank inputs and outputs. However, they can be easily generalized: permutation still reduces the number of components when (1) an orbit is not connected due to the symmetry group comprising multiple connected components, (2) the orbit does not reside on the same connected component of the minimum, and (3) there exists a permutation that takes a point on one connected component of the group to another.

---

### Official Review · Reviewer_jnZq · 2025-04-03

**Overall Recommendation:** 4

**Summary:**

This paper look at the topology of the loss level sets in order to understand their connectedness, i.e. mode connectivity. They are specifically investigating the topology of symmetry groups of the weights under the loss. They deduce the number of connected components of full-rank multi-layer linear networks (both with and without skip connections), and prove mode connectivity up to permutation for full-rank linear regressions, as well as derive some instructive examples.

**Claims And Evidence:**

It is a very sensible approach to use topology to understand the linear connectedness of minima. In terms of linear networks, the results are reasonable and informative. I appreciate the two layer examples and the examples of failure cases going from linear networks to more realistic examples. Perhaps also a more high-level discussion about the limitations of the work would be helpful--what are the main difficulties with non-linearities and how do you expect it to affect the results? Looking for some intuition.

**Essential References Not Discussed:**

I am not aware of any additional critical references beyond those mentioned, though, as stated, a broader selection of topological approaches in ML might enrich the discussion.

**Experimental Designs Or Analyses:**

The empirical results appear reasonable and align with the paper’s theoretical statements. I did not inspect the implementation details, so I cannot comment on reproducibility or correctness of the code.

**Methods And Evaluation Criteria:**

Yes, it does. I think the paper is quite dense with a fair amount of math. I would appreciate more examples and plots. Especially for an 8 page conference paper, I think that would be helpful.

**Other Comments Or Suggestions:**

N/A

**Other Strengths And Weaknesses:**

I think this is a solid paper. Topology is a brilliant and obvious tool to use study connectedness of spaces. I would appreciate some more high-level intuition of how this might generalize to more realistic NNs, as well as more examples and plots in general. However, I have not checked the proofs nor am I confident in addressing the novelty as I am not well-versed in this area.

**Questions For Authors:**

Please see comments above.

**Relation To Broader Scientific Literature:**

It looks OK to me, but I am not very familiar with related work. I would appreciate some more references to applied topology within machine learning. Now the related work only focuses on mode connectivity, but topology has been applied to ML problems and to weights of neural networks for some time, which I think deserves brief mentioning, e.g. "Exposition and Interpretation of the Topology of Neural Networks (2019)".

**Theoretical Claims:**

I have not gone through the formal verification of each proof, so I cannot assess their correctness in detail. However, the paper’s theoretical approach appears internally consistent and is built on standard techniques from topology and group theory.

---

### Decision · Program_Chairs · 2025-05-01

**Decision:**

Accept (poster)

**Comment:**

The paper presents a careful, thorough, and rigorous study of the connectivity of minima of (certain simplified) neural networks under the symmetries in the parameter space. It presents both theoretical results and some empirical evidence. The reviewers recognize the novelty and the importance of the insights gained by the work. A number of questions were brought up in the reviews, which the authors carefully addressed in the rebuttals. The primary reservation (most clearly articulated by the reviewer recommending a "weak reject", but mentioned by others) is the limited setting of the analyzed models, which have been simplified to allow for rigorous theoretical analysis. But as one of the reviewers points out, it's precisely this simplified setting that allows for proving non-trivial results, and the paper stands as a useful theoretical step in the broader program of understanding the geometry of loss landscapes. The reviewers also appreciate the honesty of scope and clarity of presentation.